# Structures of a FtsZ single protofilament and a double-helical tube in complex with a monobody

Junso Fujita [1,2,3], Hiroshi Amesaka [4], Takuya Yoshizawa[5], Kota Hibino[5], Natsuki Kamimura[5], Natsuko Kuroda[5], Takamoto Konishi[5], Yuki Kato[5], Mizuho Hara[4], Tsuyoshi Inoue [3,6,7], Keiichi Namba [1,2,8], Shun-ichi Tanaka [4] ✉ & Hiroyoshi Matsumura [5] ✉

FtsZ polymerizes into protofilaments to form the Z-ring that acts as a scaffold for accessory proteins during cell division. Structures of FtsZ have been previously solved, but detailed mechanistic insights are lacking. Here, we determine the cryoEM structure of a single protofilament of FtsZ from *Klebsiella pneumoniae* (KpFtsZ) in a polymerization-preferred conformation. We also develop a monobody (Mb) that binds to KpFtsZ and FtsZ from *Escherichia coli* without affecting their GTPase activity. Crystal structures of the FtsZ–Mb complexes reveal the Mb binding mode, while addition of Mb in vivo inhibits cell division. A cryoEM structure of a double-helical tube of KpFtsZ–Mb at 2.7 Å resolution shows two parallel protofilaments. Our present study highlights the physiological roles of the conformational changes of FtsZ in treadmilling that regulate cell division.

FtsZ is a tubulin homolog GTPase protein that is widely conserved in bacteria and plays a key role in a complex called divisome, which carries out cell division[1–3]. With GTP, FtsZ polymerizes into protofilaments and a ring-shaped structure (Z-ring), which is located in the middle of the cell and recruits more than 30 partner proteins[2,4,5]. FtsZ is tethered to the cell membrane through other proteins such as FtsA and ZipA[6–8], both of which recognize a flexible linker and C-terminal peptide (CTP) of FtsZ and stabilize highly curved FtsZ protofilaments in vitro[9,10]. FtsZ treadmilling, a motion in which the protofilament polymerizes in one end (plus-end) and depolymerizes in the other end (minus-end), drives cell wall synthesis that constricts the Z-ring to form a septum of the cell[11–14]. Although FtsZ treadmilling is associated with its GTPase activity[15,16], which is also coupled with FtsZ polymerization because the active site is formed between two FtsZ monomers[17], the

mechanism of FtsZ polymerization and depolymerization has not completely been understood.

To reveal the molecular mechanism through which FtsZ polymerizes and depolymerizes, various crystal structures of FtsZ from different species have been determined[18–21]. All of them show similar monomer conformations termed closed or relaxed (hereafter referred to as "R") and most of them do not form protofilament-like structures in the crystals. FtsZ from *Staphylococcus aureus* (SaFtsZ) was the first one to exhibit a polymerization-preferred conformation termed open or tense (hereafter referred to as "T") in the crystal structures, which all showed a protofilament-like array of the molecules[22,23]. This molecular packing with T conformation of SaFtsZ has been considered favorable for the GTPase activity because many important protein residues are closely coordinated to GTP by much closer intermolecular interactions

[1]Graduate School of Frontier Biosciences, Osaka University, 1-3 Yamadaoka, Suita, Osaka 565-0871, Japan. [2]JEOL YOKOGUSHI Research Alliance Laboratories, Osaka University, 1-3 Yamadaoka, Suita, Osaka 565-0871, Japan. [3]Graduate School of Pharmaceutical Sciences, Osaka University, 1-6 Yamadaoka, Suita, Osaka 565-0871, Japan. [4]Graduate School of Life and Environmental Science, Kyoto Prefectural University, 1-5 Hangi-cho, Shimogamo, Sakyo-ku, Kyoto 606-8522, Japan. [5]Department of Biotechnology, College of Life Sciences, Ritsumeikan University, 1-1-1 Noji-higashi, Kusatsu, Shiga 525-8577, Japan. [6]Open and Transdisciplinary Research Initiatives, Osaka University, 2-8 Yamadaoka, Suita, Osaka 565-0871, Japan. [7]dotAqua Inc., 2-1 Yamadaoka, Suita, Osaka, Japan. [8]RIKEN Center for Biosystems Dynamics Research and SPring-8 Center, 1-3 Yamadaoka, Suita, Osaka 565-0871, Japan. ✉e-mail: stanaka1@kpu.ac.jp; h-matsu@fc.ritsumei.ac.jp

with the largest interface area between two FtsZ molecules. Since then, both the T and R conformations in the filamentous structures have been revealed in many crystal structures[24–27]. Recent studies of FtsZ from two closely related species, *Escherichia coli* (EcFtsZ) and *Klebsiella pneumoniae* (KpFtsZ), have also revealed the pseudo-filamentous structures in the R conformation[28,29]. A series of studies have revealed that the "cytomotive switch" model, where the conformation switch between the T and R states is induced by the polymerization and depolymerization and not by the changes of nucleotide states, is required for treadmilling and polarity of the protofilaments[25,30,31]. As the molecular packing that gives rise to crystal structures can have confounding effects on the molecular interactions, trials have been conducted to determine the structure of FtsZ protofilaments in solution. Szwedziak et al. used cryo-electron tomography to observe reconstituted Z-rings in constricting liposomes[32], and Wagstaff et al. reconstructed a low-resolution density map of EcFtsZ protofilaments adopting the T conformation using cryo-electron microscopy (cryoEM) single particle analysis[25]. In addition, FtsZ forms mini-rings and thick helical tubes in the presence of specific support layers or accessory proteins[10,33–37]. However, due to the lack of high-resolution solution structures of the FtsZ protofilament, the relationships between these different types of polymers, the monomeric T/R conformations, and the bound nucleotides remain to be elucidated.

To answer this question, we observed KpFtsZ filaments in various nucleotide conditions and identified two types of polymerization states: thin protofilaments and thick and flexible tubes with a diameter of ~25 nm, which is very similar to the helical tubes formed by other FtsZs[33,34,36]. We determined the solution structure of a single straight protofilament of KpFtsZ by cryoEM, in which KpFtsZ was in the T conformation with GMPCPP bound. In order to aid high-resolution structure determination of the FtsZ protofilament, we developed a monobody (Mb) that binds to both KpFtsZ and EcFtsZ with high affinity. In *E. coli* cells, EcFtsZ complexed with this Mb could form Z-rings, but inhibited proper cell division, causing the cells to elongate. We determined the crystal structures of the KpFtsZ–Mb and EcFtsZ–Mb complexes at a resolution range of 2.6–1.8 Å in two different space groups, in which FtsZ was in the R conformation but the two protofilaments exhibited different curvatures. Surprisingly, the addition of the Mb stabilized the thick helical tube of KpFtsZ and enabled the structure determination by cryoEM at 2.67 Å resolution. The structure showed that two parallel protofilaments of KpFtsZ stabilized by the Mb formed a double-helical tube, in which KpFtsZ was in the R conformation. The curvature of the protofilaments in the tube was much larger than those in the cryoEM single protofilament and the crystal structures. Our present study supports the cytomotive switch model and highlights the physiological roles of the T and R conformations in treadmilling and divisome formation as well as the usefulness of Mb as a structural stabilizer.

## Results

### Observation of various types of KpFtsZ polymers in solution

To investigate the effects of nucleotides on FtsZ polymer formation, we observed KpFtsZ polymers by negative staining EM under various conditions. In the presence of GTP or GMPCPP, a slowly hydrolyzable GTP analog, very thin and bundled filaments were observed (Fig. 1a, b), as observed in several previous studies[25,38–40]. Most filaments were almost straight, but they seemed to have some flexibility. KpFtsZ alone formed thick and flexible tubes at a concentration of 1.0 mg ml⁻¹, as observed previously[33,34,36,37] (Fig. 1c). The addition of GDP or GMPPNP, a non-hydrolyzable GTP analog, did not significantly affect the tube shape (Fig. 1d, e). These tubes appeared fragile, and many nicks were observed.

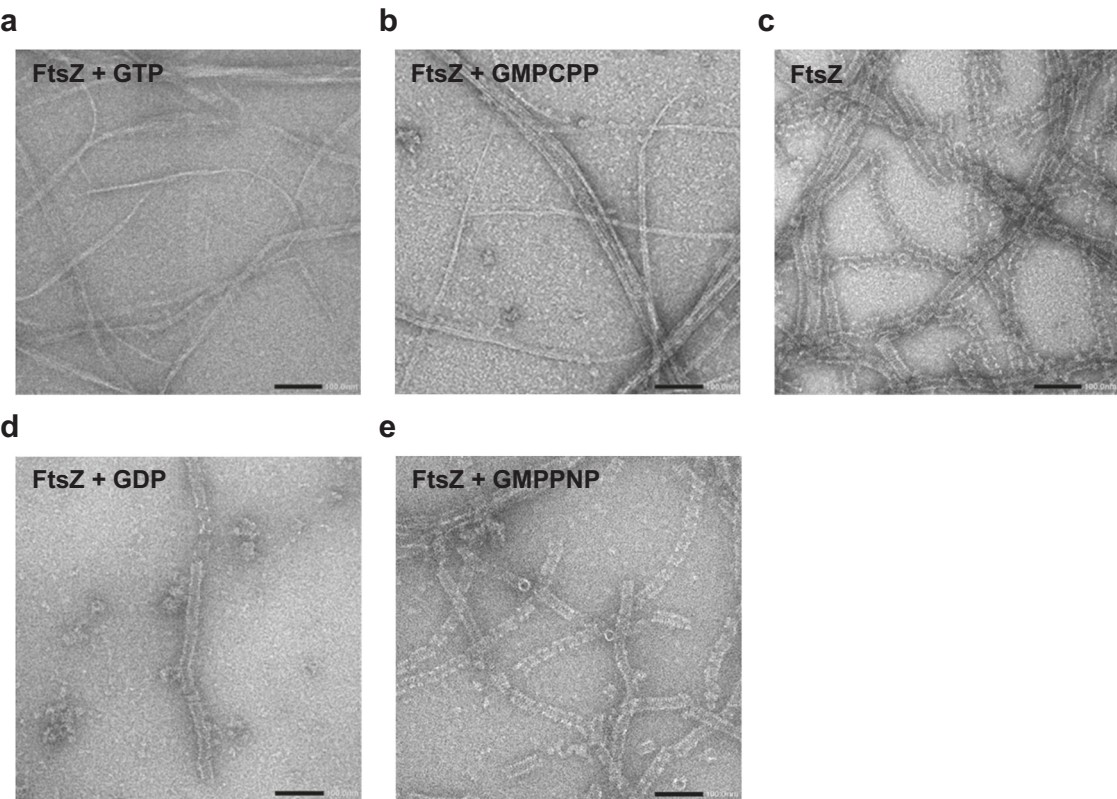

**Fig. 1 | Various morphologies of KpFtsZ filaments.** Typical negative stain micrographs of 1.0 mg ml⁻¹ FtsZ supplemented with (**a**) 1 mM GTP, (**b**) 1 mM GMPCPP, (**c**) nothing, (**d**) 1 mM GDP, and (**e**) 1 mM GMPPNP. The scale bar represents 100 nm. Each experiment was repeated independently at least twice with similar results.

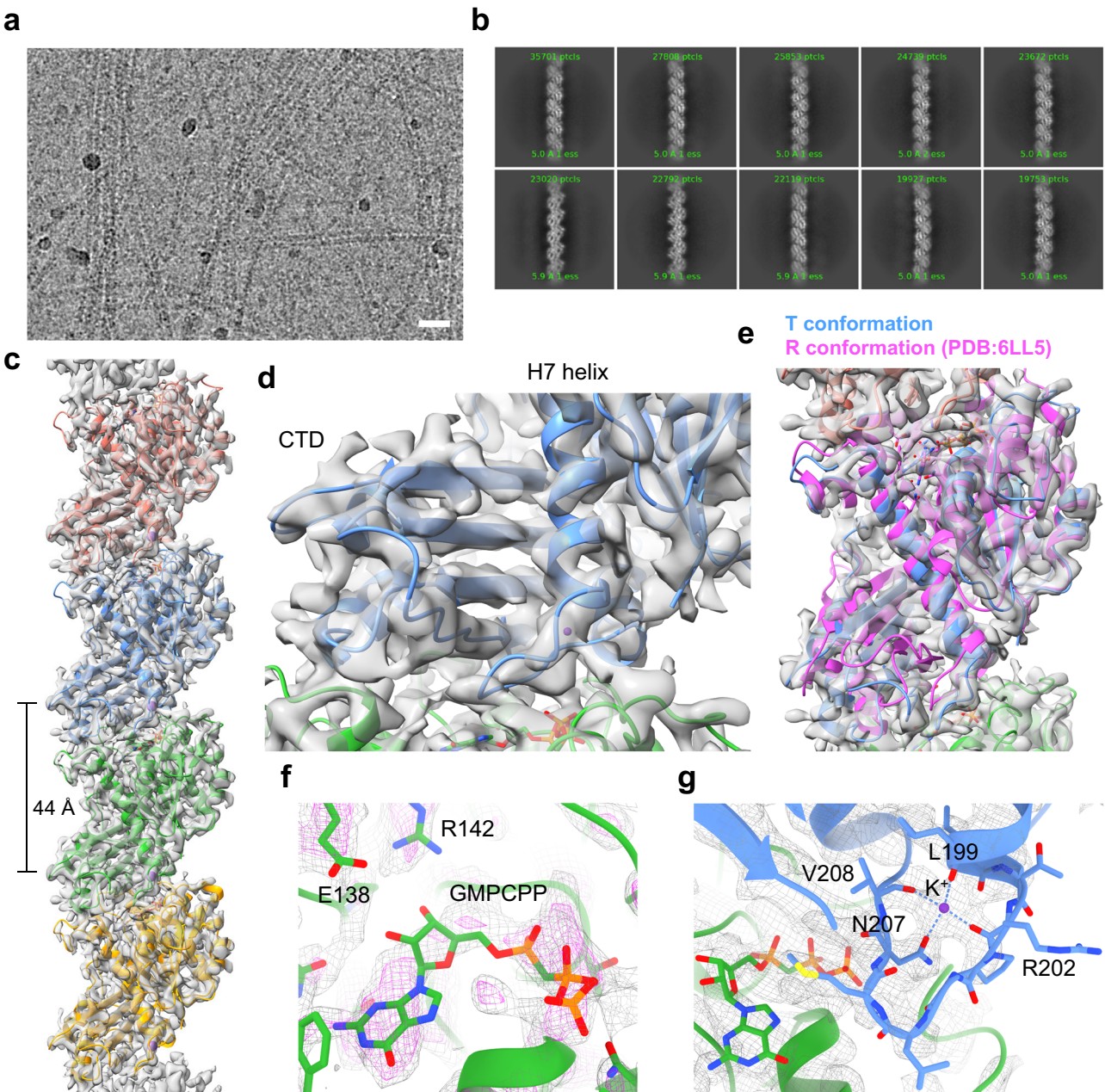

**Fig. 2 | CryoEM analysis of the KpFtsZ single protofilament. a** Typical micrograph (60,000x) of KpFtsZ supplemented with 1 mM GMPCPP. The scale bar represents 20 nm. Four grids were prepared to obtain similar results. **b** Part of 2D class averages selected for further image analysis. **c** Final sharpened map and the fitted model. Each FtsZ molecule is shown in a different color. **d** Close-up view around CTD and the H7 helix. **e** Superposition with the crystal structure of KpFtsZ in the R conformation. Close-up views around (**f**) GMPCPP and (**g**) the T7 loop and interface between two monomers. The map contour levels are 0.2 for (**c**), (**d**), and (**e**) and 0.25 for (**f**) and (**g**). The contour level of the magenta map in (**f**) is 0.35.

## CryoEM structure of the KpFtsZ single protofilament

We prepared cryoEM specimen grids for the thin filaments with 0.5 mg ml$^{-1}$ KpFtsZ supplemented with 1 mM GMPCPP. We observed many thin filaments (Fig. 2a), and the selected 2D class averages showed FtsZ monomers aligned straight with no significant twisting (Fig. 2b). After refining the helical parameters, the reconstituted 3D map represented a single protofilament of KpFtsZ with a helical rise of 44.02 Å and a helical twist of 0.025° (Fig. 2c; Supplementary Table 1). Although the resolution of the final reconstructed 3D map reached 3.03 Å (Supplementary Figs. 1 and 2a; Supplementary Table 1), the quality of the map was not enough to trace the main chain in some regions and to visualize many side chains (Supplementary Fig. 2b), and the resolution estimated from the atomic model ($d_{\text{model}}$) was limited to

3.5 Å (Supplementary Table 1). This is probably because of the biased particle orientation distribution and low sphericity derived from the flexible filamentous structure without helical twist (Supplementary Fig. 2c, d). However, the map enabled us to identify the β-sheet in the C-terminal domain (CTD) and the H7 helix (Fig. 2d; Supplementary Fig. 2e) and to confirm that KpFtsZ adopted the T conformation (Fig. 2e). GMPCPP was present in the nucleotide-binding pocket where a clear density of γ-phosphate was observed (Fig. 2f), but the density was not clear enough to determine the accurate position and conformation of the GMPCPP. Here, we put GMPCPP into one plausible position based on the strong density that is likely corresponding to the three phosphates (Fig. 2f, shown in magenta). The density corresponding to the T7 loop, which contributes to GTP hydrolysis, was also

not resolved well but bound a metal ion (Fig. 2g). The main chain carbonyls of Leu199, Arg202, Val208 and the side chain of Asn207 were coordinated to the metal ion, as is often seen in the T conformation crystal structures. Although the map resolution is not high enough to determine the accurate coordination distances, we placed a potassium ion as it is abundant in the buffer and has been seen in the previous crystal structure of SaFtsZ[27].

The KpFtsZ structure in the single protofilament is nearly identical to the T conformation of SaFtsZ[24]. When compared with the crystal structure of GTP-bound mimicking SaFtsZ complexed with GDP, BeF$_3^-$, and Mg$^{2+}$[27] (PDB code: 7OHK), both the monomer and protofilament structures are superimposed well (r.m.s.d. = 0.962 Å over 253 C$_\alpha$ atoms for the monomers) (Supplementary Fig. 3a, b). However, in the GTP-bound mimicking crystal structure of SaFtsZ (and all other FtsZ crystal structures with bound GTP or its analogs), the γ-phosphate (here BeF$_3^-$) is located near the T3 loop (Supplementary Fig. 3a, inset). In contrast, the γ-phosphate of GMPCPP in the cryoEM single protofilament structure was directed toward the front, thereby approaching the T7 loop (Fig. 2g). These differences in the nucleotide-binding mode are partially caused by switching of the hydrogen bonding partner of the ribose moiety of GMPCPP from Glu138 to Arg142 (Supplementary Fig. 3a, inset).

## Selection and characterization of a monobody targeted to KpFtsZ and EcFtsZ

To improve the map resolution and quality of the KpFtsZ protofilament, we tried to use a monobody as a small but resolvable ligand to FtsZ. To generate monobodies that bind to KpFtsZ and EcFtsZ, we selected them against EcFtsZ using two combinatorial phage display libraries: the loop library and the side library. The loop library diversifies residues in three surface loops (BC-, DE-, and FG-loops) at one end of the FN3 scaffold (Supplementary Fig. 4) and the side library diversifies residues in two β-strands (C- and D-strands) in addition to the CD- and FG-loops (Fig. 3a). We used EcFtsZ as a target for this selection as EcFtsZ and KpFtsZ are highly homologous with a sequence identity of 98.7% (Fig. 3b). After four rounds of selection against EcFtsZ, five isolates exhibiting specific binding to both KpFtsZ and EcFtsZ, as confirmed by ELISA (Supplementary Fig. 5a), were selected for sequencing. In these reads, all isolates had identical sequences derived from the side library (Fig. 3c), and we named it Mb(Ec/KpFtsZ_S1), hereafter abbreviated as Mb for brevity. Measurements of the dissociation constant ($K_d$) using surface plasmon resonance (SPR) and yeast surface display revealed that the Mb had $K_d$ values in a low μM range to KpFtsZ and EcFtsZ. The $K_d$ values measured by SPR were 24 μM for KpFtsZ and 20 μM for EcFtsZ (Fig. 3d), and those measured by yeast surface display were 5.40 ± 0.04 μM for KpFtsZ and 1.21 ± 0.34 μM for EcFtsZ (Supplementary Fig. 5b). We then tested whether the addition of different types of nucleotides affects the binding of the Mb in the yeast surface display format. The results showed that the binding did not change significantly in response to any of the nucleotides/nucleotide-analogs tested (Fig. 3e), and the Mb binding to truncated KpFtsZ (residues 11–316; KpFtsZtr) was also confirmed by gel filtration chromatography (Supplementary Fig. 5c, d), suggesting that the Mb binding site is located in the GTPase domain but not overlapped with the GTP binding site.

Next, we investigated whether the Mb binding affects the GTPase activity of KpFtsZ. The GTPase activity assay was performed using the enzyme coupling method[41], where GTP hydrolysis is coupled with a decrease in absorbance at 340 nm. As KpFtsZ supplemented with the Mb showed a slightly decreased but comparable activity to KpFtsZ alone (Fig. 3f), the Mb binding had little effect on the GTPase activity of KpFtsZ. We also observed *E. coli* cells overproducing the Mb by fluorescence microscopy to evaluate the effects of the Mb on the localization and formation of Z-rings. Mbs are cysteine-free and are therefore functional in a reducing environment inside the cells. When

EGFP was fused at the C-terminus of the Mb, the Mb-EGFP-expressing cells showed an elongated morphology compared to cells expressing EGFP alone (Fig. 3g), indicating that the Mb inhibits cell division. We measured the cell length distributions of *E. coli* overexpressing EGFP and Mb-EGFP and found that the Mb-EGFP-expressing cells were longer by ~2.7 μm on average compared to the EGFP-expressing cells (Fig. 3h). We also observed the cells co-expressing FtsZ-mCherry/Mb-EGFP or FtsZ-mCherry/EGFP. The Mb-EGFP dots were observed every 2–5 μm and overlapped with FtsZ-mCherry dots (Supplementary Fig. 6), suggesting that Mb-EGFP was bound to the FtsZ protofilaments forming Z-rings. In contrast, EGFP alone was distributed throughout the cell. Thus, it is likely that the Mb does not affect the formation of the endogenic Z-ring but inhibits its constriction.

## Crystal structures of the KpFtsZ–Mb and EcFtsZ–Mb complexes

Two crystal forms of KpFtsZtr complexed with the Mb and two crystal forms of truncated EcFtsZ (residues 11–316; EcFtsZtr) complexed with the Mb were obtained under the same crystallization conditions. The structures of the two crystal forms ($P2_12_12_1$ and $P2_1$) of the KpFtsZtr–Mb and EcFtsZtr–Mb complexes were determined using molecular replacement with the KpFtsZtr crystal structure (Protein Data Bank (PDB) code: 6LL5[28]) as the reference, at resolutions of 2.20 and 2.50 (KpFtsZtr–Mb) and 1.84 and 2.60 Å (EcFtsZtr–Mb), respectively (Supplementary Table 2). FtsZ in the $P2_12_12_1$ and $P2_1$ crystals of the KpFtsZtr–Mb and EcFtsZtr–Mb complexes all showed the R conformation, and the crystals contained one and three complex molecules per asymmetric unit, respectively. As the $P2_12_12_1$ and $P2_1$ crystal structures of KpFtsZtr–Mb were very similar to those of EcFtsZtr–Mb (a root-mean-square deviation (r.m.s.d.) of 0.245 Å for 366 C$_\alpha$ atoms for the $P2_12_12_1$ datasets and 0.341 Å for 340 C$_\alpha$ atoms for the $P2_1$ datasets; Supplementary Fig. 7a, b), we hereafter describe the crystal structures of KpFtsZtr–Mb.

The monomeric structures of Mb-bound KpFtsZtr in the two space groups were similar (r.m.s.d. of 0.453 Å for 265 C$_\alpha$ atoms; Fig. 4a). In contrast, the monomeric structures of Mb-bound KpFtsZtr and Mb-free KpFtsZtr (PDB code: 6LL5[28]) showed a relatively large difference (r.m.s.d. of 1.005 Å for 272 C$_\alpha$ atoms for the $P2_12_12_1$ dataset and 1.085 Å for 285 C$_\alpha$ atoms for the $P2_1$ dataset). The differences between these structures were observed mainly in the T3 loop and KpFtsZtr–Mb interface region. The structure of the T3 loop in KpFtsZtr–Mb showed a closed conformation similar to that of FtsZ from *Pseudomonas aeruginosa* (Fig. 4b), whereas the T3 loop in Mb-free KpFtsZtr adopted a unique open conformation probably due to the crystal packing[28]. In the KpFtsZtr–Mb complex, the Mb interacted with the central helix (H7 helix), the H6-H7 loop, and a β-sheet in the CTD. The interaction between FtsZ and the Mb was mainly mediated by Arg residues in the FG loop of the Mb (Figs. 3c and 4c; Supplementary Fig. 7c). The binding region of Mb was distant from the GTP-binding site of FtsZ, which is consistent with the finding that the GTPase activity of KpFtsZ was not mostly affected by Mb binding.

The $P2_12_12_1$ crystal structures of KpFtsZtr–Mb showed straight protofilaments in the crystal (Supplementary Fig. 8). The molecular architecture of the protofilaments was very similar to that of KpFtsZ (PDB code: 6LL5[28]), although the relative orientation of the molecules is slightly different. In contrast, the $P2_1$ KpFtsZtr–Mb structures demonstrated that the three KpFtsZtr molecules in the asymmetric unit formed a curved protofilament. The superposition of the C$_\alpha$ atoms between the $P2_12_12_1$ and $P2_1$ KpFtsZtr protofilaments composed of three KpFtsZtr molecules indicates that the protofilament of $P2_1$ KpFtsZtr is curved. The GDP portion of the middle and bottom KpFtsZtr molecules was well ordered and exhibited relatively low temperature factors (the average temperature factor is 45.2 and 50.1 Å$^2$ for GDP), compared to the average temperature factor 81.2 Å$^2$ for the GDP of the top KpFtsZtr molecule (Fig. 4d). The crystal packing

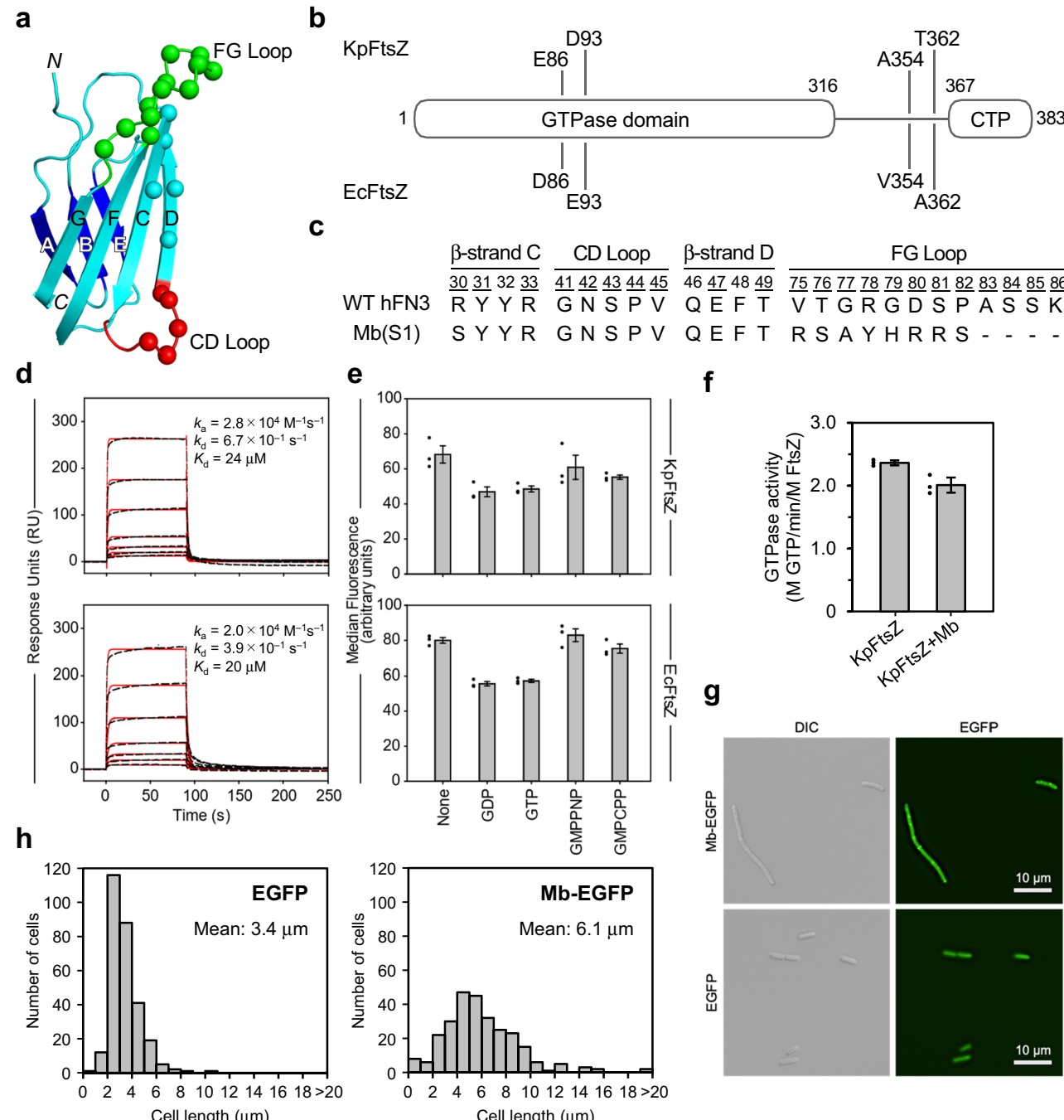

**Fig. 3 | Monobody binding to KpFtsZ and EcFtsZ. a** Schematic of the monobody scaffold, with the locations of diversified residues in the side library shown as spheres and strands, loops, and termini labeled. **b** High sequence homology between EcFtsZ and KpFtsZ. Positions with different amino acid residues between the two FtsZs are indicated. **c** Amino acid sequence of Mb, with the wild-type hFN3 sequence as a reference. Residue numbers for diversified positions are underlined. **d** SPR sensorgrams for Mb binding to KpFtsZ (upper panel) and EcFtsZ (lower panel) with kinetic parameters calculated from the best fit of a 1:1 binding model (red solid line) to the raw data (black dashed line). **e** Effect of nucleotide/nucleotide-analogs on Mb binding to KpFtsZ (upper panel) or EcFtsZ (lower panel). **f** GTPase activity assay of KpFtsZ with or without Mb. Values are shown as the mean of three independent experiments with standard deviation. **g** Differential interference contrast (DIC) and fluorescence microscopy images of *E. coli* cells overproducing EGFP and Mb-EGFP. Each experiment was repeated independently at least three times with similar results. **h** Cell length distributions of *E. coli* overproducing EGFP and Mb-EGFP. In total, 286 and 272 cells were measured for EGFP and Mb-EGFP, respectively. Source data are provided as a Source Data file.

revealed a tight packing of trimers but fewer interactions around the GDP of the top molecules.

### CryoEM structure of the KpFtsZ−Mb double-helical tube

We investigated the effects of Mb binding on FtsZ polymer formation by negative staining EM. Unexpectedly, supplementation with the Mb significantly stabilized the tubes and changed the shape to a thicker, continuous, and straight form (Supplementary Fig. 9a). These tubes appeared to have a well-ordered periodic structure, and much fewer nicks were observed.

We prepared two cryoEM specimen grids for the thick tubes: one with 4.0 mg ml⁻¹ KpFtsZ supplemented with 1 mM GMPPNP, and the

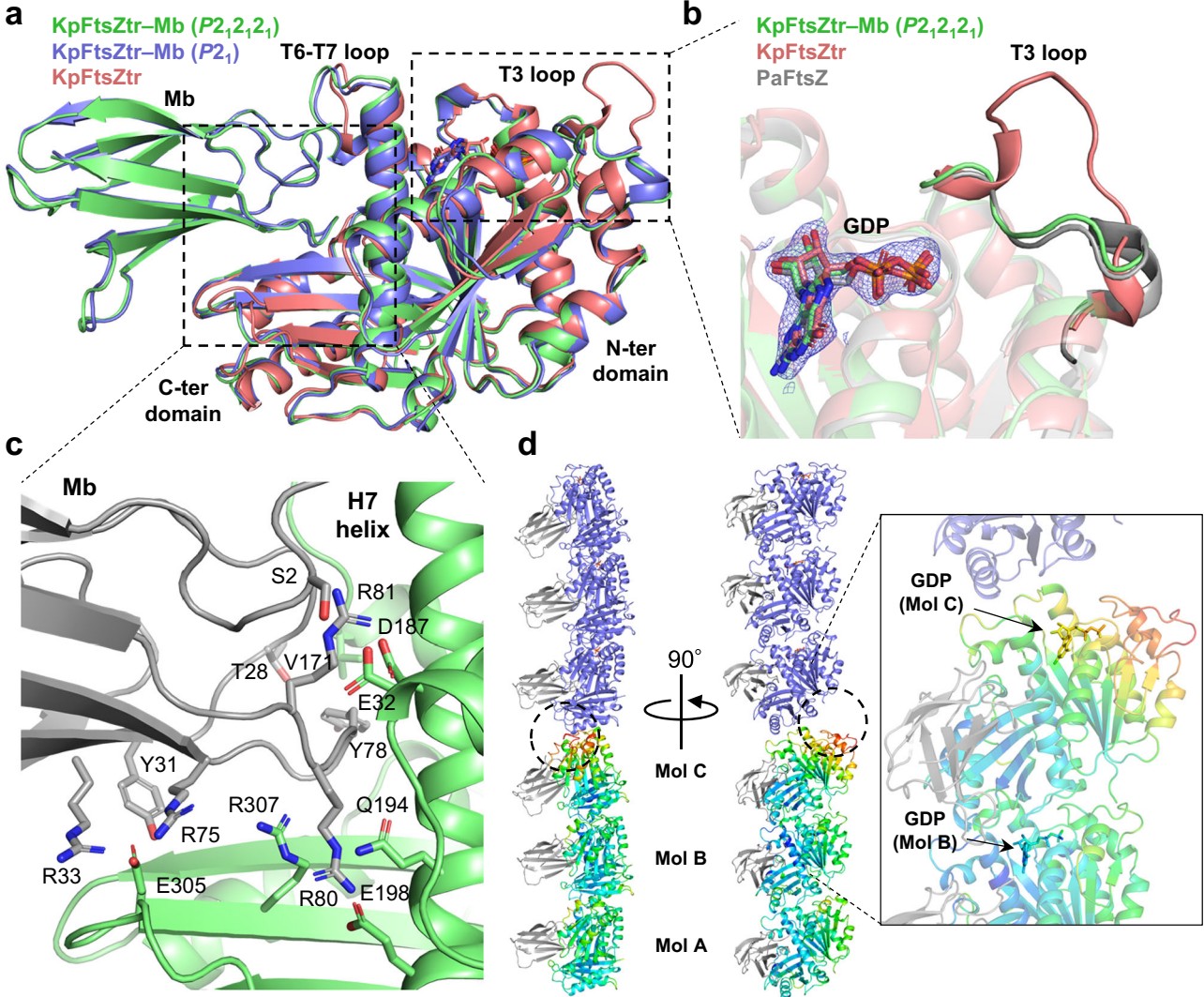

**Fig. 4 | Crystal structures of KpFtsZtr–Mb complex. a** Superimposed crystal structures of the KpFtsZtr–Mb complex in the $P2_12_12_1$ and $P2_1$ space group and Mb-free KpFtsZtr (PDB code:6LL5). Only FtsZ moiety is used for superimposition. **b** Close-up view around the T3 loop. Crystal structures of KpFtsZtr–Mb in the $P2_12_12_1$ space group, Mb-free KpFtsZtr (PDB code:6LL5), and FtsZ from *Pseudomonas aeruginosa* (PDB code:2VAW) are superimposed. The $2F_o–F_c$ map around GDP is contoured at 1.0 σ for the KpFtsZtr–Mb. **c** Close-up view around the FtsZ–Mb interface in the KpFtsZtr–Mb complex in the $P2_12_12_1$ space group. The Mb is colored gray. **d** Crystal packing of KpFtsZtr–Mb in the $P2_1$ space group. The bottom one of the pair of three FtsZ monomers in the asymmetric unit is colored by the temperature factor. The Mbs are colored gray. The inset shows a close-up view of GDP in the upper two molecules of the bottom pair.

other additionally supplemented with 1.2× molar excess of the Mb. Similar to the observation by negative staining EM, the tubes formed by KpFtsZ–Mb seemed to be straighter and more rigid than those of KpFtsZ without the Mb (Fig. 5a). After data collection and image processing, the 2D class averages of KpFtsZ–Mb showed periodic structures with a feature of helical symmetry (Fig. 5b, upper panel). In contrast, the 2D class averages of KpFtsZ showed slightly different coil spring-like structures (Fig. 5b, lower panel). We could not reconstruct a high-resolution 3D map of this coil spring structure probably because of its high flexibility, but interestingly, it showed a two-stranded helical structure, possibly indicating the nature of FtsZ to form a double-stranded protofilament in solution.

After refining the helical parameters of the KpFtsZ–Mb tube, the reconstituted 3D map revealed a strand of parallel double-helical protofilaments forming the tube with a diameter of 250 Å, with a helical rise of 7.70 Å and a helical twist of −23.40° (Fig. 5c, left panel; Supplementary Table 1). C2 symmetry along the tube axis was imposed to improve the resolution and quality of the 3D map as the two parallel KpFtsZ protofilaments formed the double helix. The Mb molecules

were located between the two protofilaments and somewhat inside the tube. We constructed a model of the KpFtsZ–Mb complex and found that each protofilament contained 15.4 molecules per turn of the helix (Fig. 5c, right panel). The resolution of the final reconstructed 3D map was 2.67 Å (Fig. 5d; Supplementary Figs. 9b, c, and 10; Supplementary Table 1).

We built a model of the KpFtsZ–Mb complex and found that the overall structure of KpFtsZ showed the R conformation. It was very similar to the crystal structures (r.m.s.d. = 0.388 Å over 366 $C_\alpha$ atoms for the $P2_1$ dataset and 0.568 Å over 360 $C_\alpha$ atoms for the $P2_12_12_1$ dataset) (Supplementary Fig. 11), and the longitudinal GDP-mediated interaction along the protofilament was maintained (Fig. 5e, see also Supplementary Movie 1). In contrast to most of the previous FtsZ structures, the N-terminal region composed of residues 3–10 was ordered (Fig. 5f). Residues 4–8 formed a short $3_{10}$ helix stabilized by the N-terminal cap of Pro4, which was located in a hydrophobic pocket formed by Glu9, Val11, Leu19, and Ser21 of the Mb of the adjacent protofilament. We found GDP in the binding pocket instead of GMPPNP (Fig. 5g), indicating that added GMPPNP did not replace GDP

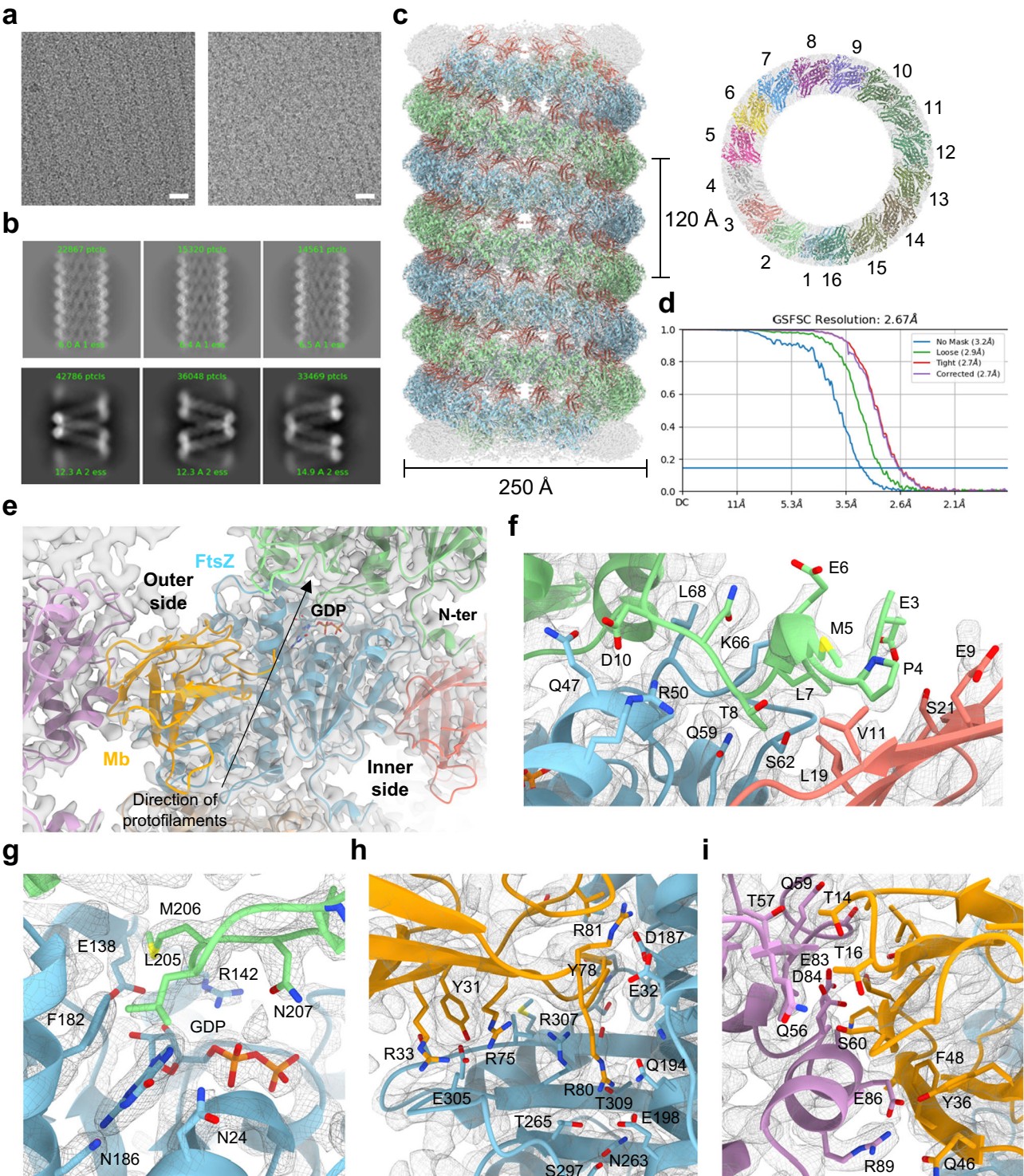

**Fig. 5 | CryoEM analysis of the KpFtsZ–Mb double-helical tube. a** Typical raw micrographs (60,000x) of KpFtsZ–Mb (left panel) and KpFtsZ (right panel). The scale bar represents 20 nm. Two grids were prepared to obtain similar results for each panel. **b** Part of 2D class averages of KpFtsZ–Mb (upper panel) and KpFtsZ (lower panel) selected for further image analysis. **c** Final sharpened map and fitted model in two orthogonal views: side view, left panel; and top view, right panel. In the left panel, two FtsZ protofilaments are shown in cyan and green. Mb molecules are shown in red. In the right panel, each FtsZ–Mb complex is shown in a different color. **d** The FSC curve for the final map. The horizontal blue line indicates the FSC = 0.143 criterion. **e** Close-up view around one FtsZ molecule (shown in cyan) and Mb (shown in yellow) fitted into the map. The molecules contributing to the longitudinal interaction (along the protofilament) are shown in green and wheat. The molecules participating in the Mb-mediated lateral interaction (between protofilaments) are shown in pink and red. Close-up view around (**f**) N-terminal interface, (**g**) GDP interface, (**h**) FtsZ–Mb interface within a complex, and (**i**) FtsZ–Mb interface between two complexes. Residues located on the surfaces are shown as sticks. The coloring is the same as (**e**), and the map is shown in mesh.

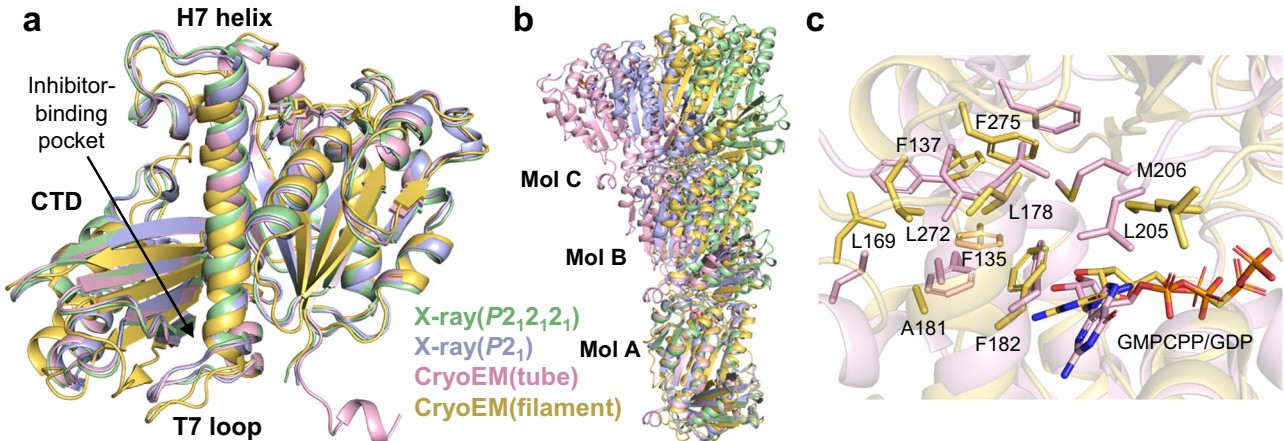

**Fig. 6 | Structural comparison of FtsZ monomers and protofilaments. a** Crystal structures of the KpFtsZtr–Mb complex in the $P2_12_12_1$ and $P2_1$ space group and cryoEM structures of the KpFtsZ–Mb double-helical tube and KpFtsZ single protofilament are superimposed. Mbs are not shown for clarity. **b** Structural comparison of KpFtsZ protofilaments of the four datasets in this study. The structures are superimposed with the bottom molecule (Mol A). **c** Close-up view of the interface between mol A and B in (**b**) in the two cryoEM datasets.

that was kept bound in KpFtsZ during the purification from *E. coli* cells. The bound GDP was sandwiched by two KpFtsZ molecules in a protofilament, similar to many previously reported crystal structures. A cluster of salt bridges and hydrophilic interactions between KpFtsZ and the Mb within the complex was maintained in the tube structure (Fig. 5h). On the other hand, the Mb contributed to lateral hydrophilic interactions between the adjacent protofilaments (Fig. 5i). Although the densities corresponding to a C-terminal region of FtsZ composed of residues 317–383 was not observed, the visible part of the C-terminal chain (up to residue 316) was located outside the tube (Supplementary Fig. 11), as previously observed[37].

### Comparison of FtsZ in the monomeric and protofilament structures

We solved four molecular structures of KpFtsZ in this study. As described above, KpFtsZ in the two crystal structures with different space groups and the cryoEM structure of the tube showed the R conformation (Supplementary Fig. 11). KpFtsZ in the cryoEM structure of the single protofilament was, however, clearly distinct from the other three and adopted the T conformation (r.m.s.d. = 1.887 Å over 280 $C_\alpha$ atoms for the $P2_1$ dataset, 1.935 Å over 279 $C_\alpha$ atoms for the $P2_12_12_1$ dataset, and 1.858 Å over 276 $C_\alpha$ atoms for the double-helical tube dataset) (Fig. 6a). The KpFtsZ monomer in the single protofilament showed the typical feature of the T conformation: the central H7 helix shifted downwards, and the CTD rotated outwards to open the inhibitor-binding pocket[26].

To evaluate the curvature of the FtsZ protofilaments, we superimposed the four models of KpFtsZ trimers generated from the two crystal structures and the two cryoEM structures with the bottom molecule (mol A) and found that the protofilament in the cryoEM double-helical tube showed the largest curvature among them (Fig. 6b). The cryoEM single protofilament showed a straight conformation similar to that of the $P2_12_12_1$ crystal structure, but the intermolecular packing was slightly different between the two. The protofilament in the $P2_1$ crystal structure represented a moderate curvature smaller than that of the double-helical tube structures. The interface areas between KpFtsZ monomers in the tube structure (1067 Å$^2$) were larger than those of the crystal structures (703 Å$^2$ for $P2_12_12_1$ and 951 Å$^2$ and 962 Å$^2$ for $P2_1$, Supplementary Table 3), suggesting that the more the protofilament is curved, the larger the interface areas become in the R conformation. The T conformation protofilament in the single protofilament structure was stabilized by larger interface areas (1114 Å$^2$) and lower free energy

($\Delta G$ = −17.8 kcal mol$^{-1}$). From a structural comparison between the T (single protofilament) and R conformations (double-helical tube) of the cryoEM structures, we found that Met206 in the T7 loop was directed to a hydrophobic core formed on top of the H7 helix in the R conformation (Fig. 6c). This non-residue-specific interaction could be one reason of protofilament stabilization in the highly curved protofilament structure formed by FtsZ in the R conformation.

## Discussion

Because FtsZ is widely conserved in bacteria and plays a key role in cell division by its GTP hydrolysis-coupled polymerization and depolymerization[1–3], the structural information of FtsZ in various polymer forms including Z-ring is essential to understand the mechanism of bacterial cell division. One of the biggest mysteries about the FtsZ function is the relationship between the different polymerization states, the T–R conformational changes, and the bound nucleotides. To answer this question, we tried to observe KpFtsZ filaments in various nucleotide conditions and distinguish the two conformations by high-resolution structure determination. Two morphologies of KpFtsZ polymers were observed in the negative staining EM images: filaments and tubes. Thin and bundled filaments were observed in the presence of GTP or GMPCPP (Fig. 1a, b), while thick tubes were observed without adding the nucleotides or in the presence of GDP or GMPPNP (Fig. 1c–e). We consider that morphologies of FtsZ polymers are strongly affected by the hydrolyzability of bound nucleotide rather than the existence of γ-phosphate. GMPCPP may be hydrolyzed by FtsZ faster than GMPPNP.

CryoEM analysis of the filaments formed in the presence of GMPCPP revealed the straight single protofilament structure formed by KpFtsZ in the T conformation with GMPCPP (Fig. 2c). This is the case not only for the solution structure of FtsZ single protofilament at the main chain resolvable resolution but also for the T conformation protofilament in the species whose crystal structure shows the R conformation in the pseudo-filamentous structures[28]. Additionally, all the crystal structures of KpFtsZ-Mb and EcFtsZ-Mb in this study adopt the R conformation regardless of either straight or curved protofilament-like structures (Supplementary Figs. 7a, b, and 8). The KpFtsZ protofilament structure ensures that the T conformation is preferable in the polymerized form and the R conformations observed in the pseudo-filamentous structures are stabilized by molecular interactions in the crystals.

The tubes were formed by FtsZ alone or by the addition of GDP or GMPPNP, but most of them were rather fragile and discontinuous

(Fig. 1c–e), and therefore cryoEM image analysis of these tubes did not produce a high-resolution 3D map. However, it revealed an interesting double-helical coil spring-like feature formed by a bundle of two protofilaments (Fig. 5b, lower panel). CryoEM structure of the tube stabilized by the Mb at 2.67 Å resolution revealed that the double-helical structure was formed by two parallel protofilaments and that KpFtsZ in the protofilament was in the R conformation with GDP bound in the nucleotide-binding site (Fig. 5c, e). We have experienced in our previous crystal structure analyses[26, 28] that without added nucleotides, FtsZ binds endogenous GDP from *E. coli* cells. The structure of our tube reveals that, despite the addition of GMPPNP in the solution, endogenous GDP was not replaced (Fig. 5g). This may be because GMPPNP's affinity for FtsZ is lower than GDP's. The intermolecular interactions along the protofilament were similar to those in the crystal structures, but the curvature of the protofilament was markedly larger than any of the crystal structures including the $P2_1$ crystal determined in the present study (Fig. 6b). We consider that the lateral interactions mediated by many Mbs stabilize the larger curvature of the protofilament in the tube.

So many crystal structures of FtsZ from different species have revealed that it can be in either of the T and R conformations regardless of the bound nucleotide and the curvature of the protofilament-like molecular array, either straight or slightly curved[19–21,24,25]. Now we confirm that the solution structure of KpFtsZ single protofilament adopts the T conformation in the presence of GMPCPP, in contrast to the crystal structures. Our results may support the cytomotive switch model, where the conformation switch between the T and R states is induced by the intermolecular interactions during polymerization and depolymerization and not by the changes in nucleotide states[25,30,31]. The ambiguity of the cryoEM map corresponding to the GMPCPP and the T7 loop might indicate that they are in a mixture of multiple conformations in the protofilaments with the T conformation. Conversely, the T conformation of FtsZ in a protofilament may not be strongly affected by the nucleotide states and the conformations of the T7 loop as far as FtsZ polymerizes.

Although the resolution of the cryoEM structure of the single protofilament is still limited to around 3.0–3.5 Å possibly due to its structural flexibility, making the examination of the structural mechanism of GTP hydrolysis reaction difficult, the present model would be quite useful as a starting model for molecular dynamics simulation to investigate it. The Mb we generated in the present study is the ligand to bind to the cavity between the central H7 helix, the H6-H7 loop, and the CTD of FtsZ, which highlights the possible binding region for inhibitors, natural ligands, and accessory proteins. Moreover, this demonstrates the usefulness of Mb, which bears the binding affinity to multiple nucleotide states of FtsZ and non-impeditive molecular size (~10 kDa) as a structure stabilizer to help determine its high-resolution structures by cryoEM as well as crystallography. We hope our present study advances our understanding of the molecular mechanisms of GTP hydrolysis and FtsZ treadmilling that regulate cell division.

## Methods

### Protein expression and purification
We used full-length and truncated versions of KpFtsZ and EcFtsZ in this study. The FtsZ derivatives were prepared as a fusion protein C-terminal to His$_6$ and a TEV cleavage site or another fusion protein C-terminal to His$_6$, a biotin-acceptor tag (Avi-tag), and a TEV cleavage site using a modified pCold I vector (Takara Bio Inc.). Expression vectors for the truncated version of KpFtsZ and EcFtsZ were constructed in our previous study[28]. For the full-length version of KpFtsZ, the cDNA of the full-length KpFtsZ gene flanked with NdeI/BamHI restriction sites for cloning (Supplementary Table 4) was synthesized by IDT (Integrated DNA Technologies, Coralville, Iowa), and the NdeI/BamHI-digested gene was inserted into the corresponding sites in the

aforementioned pCold I-derived vector. Based on the resultant vector, nucleotides 235 to 1107 of the full-length KpFtsZ gene was replaced by the corresponding gene fragment of EcFtsZ containing four mutation sites (Supplementary Table 4) using In-Fusion HD Cloning Kit (Takara Bio Inc.), yielding an expression vector for the full-length EcFtsZ. The DNA sequences were confirmed using pCold-F and pCold-R primers (Supplementary Table 5). FtsZ proteins were produced as previously described[28], except that the biotinylated FtsZs were produced in BL21(DE3) cells containing the pBirAcm plasmid (Avidity) in the presence of 50 µM D-biotin for in vivo biotinylation. The *E. coli* cells overexpressed FtsZ derivatives were harvested at 4 °C, washed once with buffer A (50 mM Tris-HCl pH 7.5, 300 mM NaCl), and broken by ultrasonication on ice. After ultracentrifugation, the soluble fraction was applied to a 5 ml HisTrap HP column (Cytiva). Elution was carried out with a gradient of eluted with 45–350 mM imidazole in buffer A. The eluted proteins were dialyzed overnight against buffer B (50 mM Tris-HCl pH 7.5) and bound to a HiTrap Q HP column (Cytiva), followed by elution in a gradient of eluted with 0–1 M NaCl in buffer B. Peak fractions were pooled and concentrated on a Vivaspin 20 (MWCO; 10,000, Sartorius) to 4 ml and loaded onto a HiLoad 16/600 Superdex200 size-exclusion column (120 ml, Cytiva) equilibrated in buffer C (20 mM HEPES-NaOH pH 7.5, 150 mM NaCl). A monobody gene was cloned into the pHFT2 vector by sticky-end PCR using the primer sets of FN5′–1/FN3′–1 and FN5′–2/FN3′–2, respectively. The DNA sequence was confirmed using T7P primer (Supplementary Table 5). The protein was prepared as a His$_{10}$-tagged protein as previously described[42,43]. *E. coli*-derived proteins, including the maltose binding protein (MBP), the phosphoenolpyruvate carboxylase (EcPEPC), the ribonuclease H2 (EcRNH2), and a mutant of the adenylate kinase (EcAdktm; A55C/C77S/V169C)[44] were also prepared as N-terminally His$_6$-tagged and biotinylated proteins using the pHBT vector[45] to validate off-target binding of monobodies. The cDNAs of these proteins were PCR amplified using either purchased or in-house constructed vectors and each of the specific primers (Supplementary Table 5). The DNA sequences were confirmed using T7P and T7T primers (Supplementary Table 5). These off-target proteins were produced with the same protocol for the biotinylated FtsZ proteins. The monobody and off-target proteins were purified using a 5 ml HisTrap HP column with the same buffers as for the aforementioned FtsZ proteins. These proteins were further purified with a ProteoSEC Dynamic 16/60 3–70 HR size-exclusion column (Protein Ark) or a HiLoad 16/600 Superdex200 size-exclusion column using an equilibrium buffer corresponding to each protein: 20 mM Tris-HCl pH 8.0, 150 mM NaCl for the monobody and MBP, 50 mM Tris-HCl pH 8.0, 150 mM NaCl, 10 mM L-Asp for EcPEPC, 20 mM HEPES pH 7.0, 500 mM NaCl, 5% (v/v) glycerol, 0.5 mM EDTA, 1 mM DTT for EcRNH2, and 50 mM Tris-HCl pH 8.0, 150 mM NaCl, 0.1 mM DTT, 0.1 mM Mg(OAc)$_2$ for EcAdktm. For enzyme assay, crystallization, and cryoEM experiments, the affinity tag for each protein was removed using TEV protease.

### Monobody generation
The monobody libraries used and general selection methods have been described previously[46,47]. The buffers used for binding reaction and washing were TBSB (20 mM Tris-HCl pH 7.4, containing 150 mM NaCl and 1 mg ml$^{-1}$ bovine serum albumin) and TBSBT (TBSB and 0.05% (v/v) Tween 20), respectively, for phage display selection experiments. Four rounds of phage display selection against EcFtsZ were performed. The EcFtsZ concentrations used for rounds 1–2 and 3–4 were 100 and 50 nM, respectively. Monobody-displayed phages were captured onto a biotinylated target immobilized to streptavidin-coated magnetic beads (Z5481/2, Promega) and then eluted in 0.1 M Gly-HCl, pH 2.1.

After four rounds of selection, individual clones from the enriched library were isolated on agar plates and six of them were subjected to titered phage ELISA using mouse anti-M13 monoclonal antibody

horseradish peroxidase (HRP) conjugate (SinoBiological 11973-MM05T-H, 1:10,000) to validate the binding and specificity against EcFtsZ and KpFtsZ. For phage preparation and ELISA assay, procedures were performed essentially as described previously[48]. Five clones binding to both EcFtsZ and KpFtsZ were selected, and their amino acid sequences were deduced by DNA sequencing.

Affinity of the generated monobody was determined using yeast-surface display as described previously[47], except that the buffers used for binding reaction and washing were HBSB (20 mM HEPES-NaOH pH 7.5, containing 150 mM NaCl and 1 mg ml$^{-1}$ bovine serum albumin) and HBSBT (HBSB and 0.05% (v/v) Tween 20), respectively. Yeast cells displaying a monobody were incubated with varying concentrations of EcFtsZ or KpFtsZ, washed with the buffer, and stained with appropriate fluorescently labeled secondary detection reagents, before analysis on a Muse flow cytometer (Millipore). Flow cytometry measurements were performed using Guava Muse Cell Analyzer (Luminex) equipped with the Muse Open Module Software (Muse Open Module Yellow and Red), and MuseSoft ver. 1.9.0.2 (Luminex). The expression of monobodies on the yeast cell surface were detected using mouse anti-V5 tag monoclonal antibody (ThermoFisher MA5-15253 or MBL Life Science M215-3, 1:1000), followed by labeling with goat anti-mouse IgG polyclonal antibody PerCP conjugate (BioLegend 405334, 1:100). Target binding was detedted using Streptavidin DyLight 550 conjugate (Abcam P22629, 1:100). $K_d$ values were determined from plots of the median fluorescent intensity against FtsZ concentration by fitting the 1:1 binding model using SigmaPlot ver. 15.0 (Systat Software). Experiments were performed in triplicate. The detail of the gating strategy is described in Supplementary Fig. 12.

## Negative staining

Amorphous carbon grids were hydrophilized on one side by glow discharge using a JEC-3000FC sputter coater (JEOL). 3 µl of purified KpFtsZ at a concentration of 1.0 mg ml$^{-1}$ was loaded on each grid and blotted. Then each grid was immediately stained with 3 µl of 2% uranyl acetate solution and blotted, and this process was repeated three times. Each grid was air-dried for 30 min. Specimens of KpFtsZ supplemented with any of 1 mM GDP, GMPPNP, GTP, GMPCPP, or 1.2× molar excess of Mb were prepared in the same way. Images were taken using JEM-1400Flash (JEOL, Japan) operated at 100 kV.

## CryoEM specimen preparation and data collection

For the KpFtsZ–Mb double-helical tube, the solution containing 4.0 mg ml$^{-1}$ KpFtsZ, 20 mM HEPES pH 7.5, 25 mM NaCl, 100 mM KCl, 5 mM MgCl$_2$, 1 mM GMPPNP, 0.12 mM PC190723, and 1.2× molar excess of Mb was incubated on ice for 20 min. A KpFtsZ-only sample (without the addition of PC190723 and Mb) was prepared in the same way. Epoxidized graphene grid (EG-grid)[49] was used to stabilize and increase the number of FtsZ filaments. 3 µl of the 0.01 M NaOH and 1% (v/v) epichlorohydrin water solution was applied to the ClO$_2$$^-$-oxidized graphene grids to prepare EG-grids. Then 3 µl of the protein solution was applied to the EG-grid and incubated at room temperature for 5 min. For the KpFtsZ single filament, the solution containing 0.5 mg ml$^{-1}$ KpFtsZ, 15 mM HEPES pH 7.5, 10 mM NaCl, 100 mM KCl, 5 mM MgCl$_2$, and 1 mM GMPCPP was incubated on ice for 30 min. Quantifoil grids (R1.2/1.3 Cu 200 mesh) were glow-discharged using a JEC-3000FC sputter coater (JEOL, Japan) at 20 mA for 20 s. 3 µl of the solution was applied to the glow-discharged grids in a Vitrobot Mark IV chamber (Thermo Fisher Scientific, USA) equilibrated at 4 °C and 100% humidity. The grids were blotted with a force of 0 and a time of 3 s (KpFtsZ–Mb double-helical tube) or with a force of −10 and a time of 1.5 s (KpFtsZ double-helical tube and single filament) and then immediately plunged into liquid ethane. Excess ethane was removed with filter paper, and the grids were stored in liquid nitrogen. CryoEM image datasets were acquired using SerialEM ver. 3.8 or 4.0[50], yoneo-Locr ver. 1.0[51], and JEM-Z300FSC (CRYO ARM™ 300: JEOL, Japan) or

JEM-3300 (CRYO ARM™ 300 II: JEOL, Japan) operated at 300 kV with a K3 direct electron detector (Gatan, Inc.) in CDS mode. The Ω-type in-column energy filter was operated with a slit width of 20 eV for zero-loss imaging. The nominal magnification was 60,000×, corresponding to approximately 0.88 Å per pixel. Defocus varied between −0.5 µm and −2.0 µm. Each movie was fractionated into 60 frames (0.0505 s each, total exposure: 3.04 s) with a total dose of 60 e$^-$/Å$^2$.

## CryoEM image processing and model building

The gain reference was generated from 500 movies in the dataset with the "relion_estimate_gain" program in RELION 4.0[52]. For the KpFtsZ single filament, the images were processed using cryoSPARC ver. 4.1.1[53]. 6,079 movies of the dataset were imported and motion corrected, and the contrast transfer functions (CTFs) were estimated. 5439 micrographs whose CTF max resolutions were beyond 5 Å were selected. To prepare a 2D template, 269,764 particles were automatically picked from 500 micrographs using filament tracer job with the parameters; filament diameter, 50 Å; separation distance between segments, 1. The particles were extracted with a box size of 360 pixels with 2x binning. After two rounds of 2D classification, 2D class averages of single filaments with visible secondary structures were selected as templates. 3,695,707 particles were automatically picked from all micrographs using the templates in a filament tracer job with the parameters; filament diameter, 44 Å; separation distance between segments, 1. After the images of the hole edges were removed, the particles were extracted with a box size of 360 pixels with 2x binning. After two rounds of 2D classification, 551,739 particles were selected and extracted with a box size of 360 pixels binning to 300 pixels. The extracted particles were subjected to helix refine job with an estimated helical rise of 44 Å and a helical twist of 0°. After confirming that these values are correct with symmetry search job, another round of helix refine was performed. After global and local CTF refine jobs, the final helix refine job was run to reach the map resolution of 3.03 Å (FSC = 0.143) with the optimized helical rise of 44.02 Å and the helical twist of 0.025°. The entire workflow is shown in Supplementary Fig. 1.

For the KpFtsZ–Mb double-helical tube, the images were processed using cryoSPARC ver. 3.3.1[53]. 3,096 movies of the KpFtsZ–Mb dataset were imported and motion corrected, and the contrast transfer functions (CTFs) were estimated. 2889 micrographs whose CTF max resolutions were beyond 5 Å were selected. 181,247 particles were automatically picked using filament tracer job with the parameters; filament diameter, 250 Å; separation distance between segments, 0.5; minimum filament length to consider, 3. At an early stage of processing, particles were extracted with a box size of 480 pixels with 2x binning. After several rounds of 2D classification, helix refine, and symmetry search, approximate values of helical rise and twist were found. Then particles were extracted with a box size of 600 pixels without binning. After two rounds of 2D classification, 90,711 particles were selected. The particles were subjected to helix refine job with the approximate helical parameters and without imposing symmetry. After global and local CTF refine jobs, another helix refine job was performed. Then the final helix refine job was run with C2 symmetry. The refined values of helical rise and twist were 7.70 Å and −23.40°, respectively, and the final map resolution (FSC = 0.143) reached 2.67 Å. The dataset of KpFtsZ was also processed similarly, but the helical parameters were difficult to refine due to the structural flexibility of the filaments, and the map resolution was limited to 7–8 Å. The entire workflow is shown in Supplementary Fig. 10.

The monomeric models of KpFtsZ and KpFtsZ–Mb were built using the crystal structure of KpFtsZtr (PDB code:6LL5[28]) and the $P2_12_12_1$ crystal structure of KpFtsZtr–Mb complex as initial models, respectively. After the initial model was manually fitted into the map using UCSF Chimera ver. 1.15[54] and modified in Coot ver. 0.9.6[55], real-space refinement was performed in PHENIX ver. 1.19.2[56]. The model was validated using MolProbity ver. 4.5.2[57], and this cycle was repeated

several times. Four refined monomer models were fitted into the map of the KpFtsZ single filament, and real-space refinement was performed in PHENIX again. The whole KpFtsZ–Mb tube model (containing 100 KpFtsZ–Mb complexes) was generated using the following command in UCSF Chimera: "sym #1 group C2*H,7.703,−23.398,50,−25 center 264.9,264.9,264.9". Interface areas were calculated with PISA server[58]. Figures were prepared using ImageJ ver. 1.53q[59], UCSF Chimera ver. 1.15[54], ChimeraX ver. 1.1[60], PyMOL ver. 2.5.0 (Schrödinger, LLC, USA), and LigPlot⁺ ver. 2.2.4[61]. The parameters are summarized in Supplementary Table 1.

### Surface plasmon resonance (SPR)
SPR measurements were performed in 10 mM HEPES-NaOH pH 7.4, containing 150 mM NaCl and 0.005% (w/v) Tween 20 at 25 °C on a Biacore T100 instrument with control Software ver. 1.1.1 (Cytiva). The monobody was immobilized via a histidine tag to a Series S Sensor Chip NTA (Cytiva). KpFtsZ or EcFtsZ at varying concentrations (0, 0.5, 1, 2, 4, 10, 20, and 40 μM, respectively) were flowed over the sensor chip at a rate of 20 μl min⁻¹ and the binding signal was monitored. The kinetic traces after background subtraction (with no monobody immobilized) were fitted to the 1:1 binding model using the Biacore T100 evaluation software ver. 1.1.1 (Cytiva).

### Effect of nucleotides on monobody binding
The effect of nucleotides on monobody binding was examined using yeast-surface display as described above, except that an appropriate concentration of EcFtsZ or KpFtsZ was pre-incubated with or without a nucleotide/nucleotide-analog (1 mM for GDP, GTP, and GMPPNP and 0.1 mM for GMPCPP) in HBSB for 30 min on ice, and then the mixture was transferred to the wells where monobody binding took place. The concentrations of EcFtsZ and KpFtsZ used were 1.5 μM and 10 μM, respectively, to account for different affinity. Experiments were performed in triplicate.

### Gel filtration chromatography of KpFtsZtr–Mb complex
Gel filtration chromatography of KpFtsZtr–Mb complex was carried out using a 24 ml Superdex 200 column (Cytiva) equilibrated with 20 mM HEPES-NaOH pH 7.5, 150 mM NaCl. 18.9 mg ml⁻¹ KpFtsZtr and 9.9 mg ml⁻¹ Mb were mixed and incubated at room temperature for 1 h before being subjected to gel filtration chromatography. The isolated complex had an apparent molecular mass of 49.9 kDa as estimated from gel filtration chromatography.

### GTPase activity assay
GTPase assay of KpFtsZ was performed using a coupled enzyme assay[41]. In this method, GTP hydrolysis is coupled to NADH oxidation with pyruvate kinase (PK) and lactate dehydrogenase (LDH). A decrease of NADH absorbance at 340 nm was measured, which is proportional to the rate of GTP hydrolysis. In our experiments, the sequential reactions were performed in 50 mM Tris-HCl pH 7.5, 5 mM MgCl₂, 200 mM KCl, 50 U ml⁻¹ PK, 50 U ml⁻¹ LDH, 1 mM phosphoenolpyruvate (PEP), 1 mM GTP, 0.2 mM NADH, and 0.81 mg ml⁻¹ FtsZ at 37 °C. The total reaction volume was 1 ml. The GTPase activity was calculated from the slope of the fitted line in the area with linearly decreasing absorbance at 340 nm divided by the molar extinction coefficient of NADH (6220 M⁻¹cm⁻¹), FtsZ molar concentration (20 μM), and the path length (1 cm). Data are obtained as the mean of three independent experiments with standard deviation.

### Fluorescence microscopy
For imaging experiments, we generated expression vectors for the fluorescent protein-fused constructs using In-Fusion HD Cloning Kit. Specifically, the mCherry and EGFP genes were fused to the C-terminal ends of EcFtsZ and Mb, respectively, in the aforementioned vectors.

The coding regions of these constructs are summarized in Supplementary Table 6. *E. coli* BL21(DE3) was transformed with Mb-EGFP or EGFP expressing vector and grown in LB medium until OD₆₀₀ = 0.388. The proteins were induced by the addition of 0.5 mM isopropyl β-ᴅ-thiogalactopyranoside (IPTG). 2 μl of the samples were applied to the glass slide after 25 min of induction and then covered with 18 mm × 18 mm cover glass. Cells were imaged using inverted fluorescence microscopy (Nikon, Eclipse Ti2) equipped with a 100× oil immersion objective lens (Nikon). The microscope system was operated using the NIS-elements ver. 5.21.00 (Nikon).

### Protein crystallization
Purified KpFtsZtr or EcFtsZtr at a concentration of 19 mg ml⁻¹ was mixed with 1.2× molar excess of Mb and PC190723[62] (a known FtsZ inhibitor). The final concentration of KpFtsZtr and EcFtsZtr, and Mb are 11.8 mg ml⁻¹ and 4.0 mg ml⁻¹, respectively. KpFtsZtr–Mb and EcFtsZtr-Mb crystals belonging to the space group *P*2₁ and *P*2₁2₁2₁ were obtained by hanging-drop vapor diffusion at 293 K (1 μl protein solution + 1 μl reservoir solution) with the same reservoir solution consisting of 0.2 M sodium formate, 0.1 M Bis-Tris propane pH 6.5, and 20% PEG3350.

### Crystallographic data collection, processing, and refinement
Crystals were flash-cooled in a stream of nitrogen at 100 K without cryoprotectants after mounting in a loop. X-ray diffraction data were collected at a wavelength of 0.900 Å on the micro-focus beamline BL44XU at SPring-8, Hyogo, Japan using an EIGER X 16 M detector (Dectris). The datasets for both KpFtsZtr–Mb and EcFtsZtr–Mb were integrated, and scaled using the KAMO system[63] which runs BLEND[64], XDS, and XSCALE ver. 5[65] (February 2021) automatically. The phases for each structure were determined by molecular replacement with MOLREP[66] in the CCP4 suite ver. 7.1[67] using the previously determined structure of KpFtsZtr (PDB code:6LL5[28]) as the search model. Each model was refined with REFMAC ver. 5.8.0267[68] and PHENIX ver. 1.19.1[56], with manual modification using Coot ver. 0.8.6[55]. The refined structures were validated with MolProbity ver. 4.5.2[57]. Ramachandran statistics (percentages of favored/allowed/disallowed) are 97.21/2.54/0.25 for KpFtsZtr–Mb (*P*2₁2₁2₁), 94.83/4.57/0.59 for KpFtsZtr–Mb (*P*2₁), 98.97/0.77/0.26 for EcFtsZtr–Mb (*P*2₁2₁2₁), and 92.56/5.81/1.62 for EcFtsZtr–Mb (*P*2₁). Data-collection and refinement statistics are shown in Supplementary Table 2.

## Data availability
CryoEM atomic coordinates, maps, and raw multi-frame movies have been deposited in the Protein Data Bank (PDB), the Electron Microscopy Data Bank (EMDB), and EMPIAR under the accession codes 8IBN, EMD-35344, and EMPIAR-11426 for KpFtsZ single filament, and 8H1O, EMD-34429, and EMPIAR-11425 for KpFtsZ–Mb double-helical tube, respectively. Coordinates and structure factors have been deposited in PDB under the accession numbers 8GZV (KpFtsZtr–Mb), 8GZW (KpFtsZtr–Mb), 8GZX (EcFtsZtr–Mb), and 8GZY (EcFtsZtr–Mb). The other coordinates used in this study are available from the PDB; 7OHK, 6LL5, and 2VAW). Source data are provided with this paper.

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

## Acknowledgements

This work was supported by: JSPS KAKENHI grant JP18K05445 (S-i.T.), JP21K05386 (S-i.T.), JP18K06094 (H.M.), JP19H04735 (H.M.), and JP20K22630 (J.F.); the Sasakawa Scientific Research Grant from the Japan Science Society 2018-3011 (S-i.T.), and 2022-4052 (H.A.); JST OPERA (Open Innovation with Enterprises, Research Institute and Academia) grant JPMJOP1861 (T.I., K.N.); AMED BINDS (Platform Project for Supporting Drug Discovery and Life Science Research (BINDS) grant JP21am0101117 and JP22ama121003 (K.N.), and JP21am0101070 (H.M.); AMED CiCLE (Cyclic Innovation for Clinical Empowerment) grant JP17pc0101020 (K.N.); JEOL YOKOGUSHI Research Alliance Laboratories of Osaka University (K.N.); G-7 Foundation Grant (H.M.); the Cooperative Research Program of Institute for Protein Research, Osaka University (CR-20-02 and CR-21-02). We thank Kiyoaki Arakawa, Miho Emori, Takahiro Hayashi, Momoka Iiyama, and Naoki Okazaki for help in monobody screening by phage display, and Yoshie Kushima for help in negative staining. This work has been performed under the approval of the SPring-8 Program Advisory Committee (Proposal Nos. 2020A2536, 2020A2544, 2020A6557, 2021A6623, 2021A6648, and 2021B1002).

## Author contributions

J.F., S-i.T., and H.M. designed the research. H.A., M.H., and S-i.T. acquired the monobody and analyzed its binding affinity. T.Y., K.H., N.K., N.K., T.K., Y.K., and H.M. prepared the protein samples and performed activity assay, binding assay, crystallization, crystallographic data collection, processing, and refinement. T.Y. and N.K. observed E. coli cells using fluorescent microscopy. J.F., K.H., N.K., N.K., T.K., Y.K., and H.M. performed negative staining and cryoEM data collection. J.F. performed cryoEM image processing and model building. J.F., H.A., T.Y., S-i.T., and H.M. prepared the figures and wrote the first draft of the manuscript. T.I. and K.N. helped to analyze and interpret the data, and critically revise the manuscript. J.F., S-i.T., and H.M. conceptualized the study, developed the study design, supervised the authors throughout the study, and provided expertize in manuscript preparation. All authors read and approved the reviewed manuscript.

## Competing interests

The authors declare no competing interests.
