## [Peer Review File · Nature Communications]

Structures of a FtsZ single protofilament and a double-helical tube in complex with a monobodyREVIEWER COMMENTS

Reviewer #1 (Remarks to the Author):

The authors of this study have generated a monobody (Mb) that binds to bacterial cell division protein FtsZ. They employed this Mb as a crystallization and assembly chaperone for FtsZ structural studies.

Crystal structures of KpFtsZ-Mb and EcFtsZ-Mb complexes, in two space groups each, show that the Mb binds on a large lateral cavity formed by the central helix H7, the H6-H7 loop, and a beta-sheet in the C-terminal domain. FtsZ in these complexes is in the relaxed (R) conformation, as opposed to the tense (T) conformation active for canonical protofilament assembly. This is possibly the first ligand demonstrated to bind to this FtsZ cavity, which suggests to me the possibility of targeting FtsZ with inhibitors binding to this cavity. It also raises the question of whether natural ligands or partner proteins may bind there.

In solution, the Mb induces KpFtsZ to form 25 nm diameter double helical tubes, made of filaments at an angle of 66.6 degrees with respect to the tube axis. The structure of these FtsZ-Mb tubes has been determined by cryoEM to 2.67 Å resolution. These tubes are possibly related to other FtsZ tubes and mini-rings previously reported in the literature. The geometry of FtsZ-Mb tubes may also be compared to those of curved GDP-tubulin polymers (Wang & Nogales, Nature 2005), tubulin-stathmin complexes (Gigant et al., Cell 2000) and GDP-tubulin rings (Diaz et al., J Mol Biol 1994). All of them are different from microtubules, where the protofilaments align with the microtubule axis.

The work is interesting and technically sound. The manuscript is overall clearly written and figures are informative. Nevertheless, the biological relevance of the results needs to be supported by additional experimental evidence. Detailed comments follow, wishing that some of them may be useful to the authors.

First, the "microtubule-like" claim for the FtsZ-Mb tubes in the title, abstract and throughout the manuscript is not appropriate, essentially due to distinct filament organization, as explained above. In addition, the speculation that the FtsZ-Mb tubes 'reflect the structure of the Z-ring in some phases during cell division' (Discussion) is unjustified. Moreover, the authors state that 'a constricted ring should consist of highly curved protofilaments' but a constricted ring could also arise from bundles of short coalescent filaments exhibiting minimal curvature. Bundling of short filaments in clusters, giving rise to a discontinuous Z- ring, is supported by super-resolution microscopy (Bisson-Filho et al., Science 2017; Yang et al., Science 2017; Monteiro et al., Nature 2018).

Second, the significance of R conformation GDP-FtsZ tubes for bacterial division is uncertain. There is consensus in the field that filaments exist in the T configuration, while the R configuration is mostly present in monomers. Is it possible that the FtsZ-Mb tubes showing a large curvature represent an intermediate of dissociation rather than a configuration in the Z-ring, as proposed by the authors?

Related to the previous comment, would it be possible to study T-conformation polymers, for example by adding the Mb to the FtsZ+GTP sample shown in Figure 3e? Could the authors please clarify which polymers are formed by KpFtsZ with Mb and GTP or GMPCPP? And why the functionally relevant thin polymers (Figure 3e) are excluded from the cryoEM analysis? This would provide more meaningful mechanistic insights. Besides, why add GMPPNP for cryoEM if the bound nucleotide remains GDP from the purification stage (end of page 8)? And why is PC190723, which is known to stabilize the T-conformation filaments, added to the sample used for cryo-EM studies? Third, the results shown in Figure 1g are very interesting, as they indicate that a Z-ring could be formed in the presence of the Mb. Nevertheless, a single image may not be representative and quantification of cell length is required. Importantly, in order to confirm that the Mb-EGFP dots are really Z-rings, the authors should show that FtsZ co-localizes with Mb-EGFP dots, for example by labeling FtsZ with RFP/mCherry. Moreover, Figure 1f shows that the GTPase activity is maintained in the presence of the Mb. How do the authors interpret that GTPase activity is not affected while cell division is inhibited?

Minor comments:

- Page 2, line 4. The authors claim that high resolution structures are lacking. Of what exactly? Atomic-resolution structures of FtsZ filaments in the T configuration and pseudofilaments in the R conformation have been reported.

- Page 3, line 6. The text ‘...FtsA and ZipA, ..., and C-terminal peptide of FtsZ and stabilize highly curved FtsZ filaments’ should read ‘...stabilize highly curved FtsZ filaments in vitro’.
- Page 3, line 8. The ‘treadmilling’ concept deserves a thorough description for a general readership.
- Page 4, line 18. It may be useful for the readers to explain better what are ‘the loop library and the side library’.
- Page 5, second paragraph and Figure 1f. What is the extent of FtsZ polymerization in the GTPase assays?
- Page 6, line 16 and Figure 1b are not coherent. In the text ‘C-terminal domain’ is used for the GTPase-activating domain and not for the CTD shown in the figure.
- Page 6, line 21 and Ext. Figure 5. How do these filaments compare with the reported crystal structure of KpFtsZ lacking Mb (Yoshizawa et al., Acta Cryst D 2020)?
- Page 7, line 14. Cite previous studies.
- Page 9, line 12. ‘Non-impeditive’ of what? This whole sentence is difficult to understand.
- Page 9, line 19. In Figure 6a, how is the -Mb straight model generated? In particular, how is the lateral blue-green interaction deduced? Is it supported by crystal contacts in the structures described in this or past reports?
- Page 10, second paragraph. I do not understand the meaning of the sentence ‘the shapes of the KpFtsZ filaments are not affected by the presence of gamma phosphate (GTP like or GDP like)’. I also do not understand the sentence ‘GTP hydrolysis would be required for the formation of long and straight FtsZ protofilaments’.
- Page 16, line 6. Please include a brief description of the purification method, including buffer compositions and nucleotides/additives used along the purification.
- Page 17, second last line. ‘took place’ instead of ‘took plate’.
- Page 19, line 5. In the crystal structures of FtsZ-Mb a 1:1 FtsZ:Mb complex is observed. For protein crystallization FtsZ ‘was mixed with a 1.2x molar excess of Mb’. However, ‘final concentrations of FtsZ and Mb were 8 mg/ml and 0.4 mg/ml’, which if I am not wrong is a 0.2 molar ratio of Mb to FtsZ. Is this an error?
- Table 1. Please report Ramachandran statistics.

Reviewer #2 (Remarks to the Author):

Review of NCOMMS-22-41370-T “High-resolution structure of a microtubule-like tube composed of FtsZ-monobody complexes”

1. Summary

In this article, the authors describe their investigation into the structure of an artificial filament containing the bacterial cytoskeletal protein FtsZ. This filament owes its structure to a 10 kDa monobody that the authors raised against FtsZ, which binds away from FtsZ’s GTPase region. This monobody does not inhibit Z-ring formation (or more precise FtsZ localization) in live cells, but does inhibit division. The authors solved several crystal structures and a cryo-EM structure. They identify various interfacial interactions in their maps, whose effects on the higher-order structure of FtsZ protofilaments they discuss. Despite the authors’ attempts, they did not obtain a cryo-EM structure of the filament without their monobody and they discarded the GTP/GMPCPP-bound forms of FtsZ, which are the most likely in vivo form of the filaments, because they were too thin for analysis. To me it is not clear how the monobody-bound filament relates to the biological reality, if at all. The authors cite FtsZ’s role in vivo, which requires that it adopt a range of curvatures. Of course, the monobody is absent in unmodified cells, so this study cannot answer the question of what drives FtsZ filaments to curve during cell division.

In the discussion (and the title), the authors make a comparison of their monobody-bound filaments to microtubules. Microtubules are composed of eukaryotic tubulins, to which prokaryotic FtsZ bears homology. However, as the authors point out, their tubular complex differs from microtubules in several significant ways. The direction of the protofilaments is entirely different. The authors’ complex forms a 2-start helix, while microtubule subunits form a curved sheet.

Moreover, the authors' structure is under the considerable influence of their monobody, which occurs in equal number to FtsZ monomers. In my opinion, the comparison to MTs is unwarranted and misleading and demonstrates the author's inability or unwillingness to interpret their work in the light of biological reality.

Furthermore, in their brief discussion of the tubular complex's biological relevance, the authors propose that molecules that bind FtsZ in the same manner as their monobody, could induce - according to their size - different inter-protofilament spacings during Z-ring constriction. But during constriction, the curvature of the ring will increase in a continuous fashion. Each different hypothetical crosslinker would presumably rigidly induce a particular spacing. That Z-ring constriction would be quantised like this seems highly unlikely. In fact, the necessary flexibility of the Z-ring surely supports the hypothesis that it is not under the influence of a rigidly crosslinking binding partner.

I must conclude that the author's discussion is not insightful or helpful.

While the authors' results are technically impressive, the relevance of their work to FtsZ's role in vivo is at best unclear. I suggest the work needs a completely redrafted title and discussion, and significant editorial adjustments throughout.

2. Individual points

The following points are mostly about specific uses of language, some of which the authors might find helpful.

Introduction

1) "various studies have revealed its molecular mechanisms" Perhaps "molecular structure" would be a better choice of words. To my mind, a mechanism is a process through which some action is performed. It is also not clear to me to whether "it" refers to FtsZ or the Z-ring.

2) The last part of the first sentence is poorly worded: "plays a key role in a complex called divisome that functions during cell division". The divisome doesn't just function during division - it carries out division.

3) The authors write: "FtsZ polymerizes into protofilaments and a ring-shaped structure (Z-ring) with GTP, which is located in the middle of the cell and constricts to provide force for dividing the cell in half.". It is the Z-ring, not GTP, that is located in the middle of the cell. To clear this, up might I recommend: "With GTP, FtsZ polymerizes into protofilaments and a ring-shaped structure (Z-ring), which is located in the middle of the cell and constricts to provide force for dividing the cell in half."

4) FtsZ is tethered to the cell membrane [through by] other proteins ... This first paragraph seems to contradict itself, saying initially that it is Z-ring constriction that provides the force necessary to divide the cell, and then that it is the remodelling of the cell wall. If the authors wish to communicate that it is not known from where the necessary force comes, then they should say so.

5) Rather than "the molecular mechanism leading to the FtsZ function", I suggest maybe "the molecular mechanism through which FtsZ [polymerises]".

6) The authors have: "these crystal structures cannot escape the effect of crystal packing". Crystal structures in general cannot escape the effects of crystal packing. This sentence misses the point that these effects are often undesirable. I suggest: "The crystal packing that gives rise to crystal structures can have confounding effects on molecular interactions."

7) Rather than "inhibits proper cell division to elongate the cells", how about "inhibits proper cell division, causing the cells to elongate":

8) "Our structural analyses showed the structural similarity and difference between the microtubule and FtsZ tube and the plasticity of the FtsZ protofilament, which may be important for the formation of Z-rings in various sizes for completing cell division.". 'm not certain what this sentence is intended to communicate. Also, the word "structural" is not necessary. By now, it is well established that we are talking about the structure of FtsZ filaments. Perhaps the authors would be better served by dividing this sentence. "We see ways in which the FtsZ tube is similar to microtubules and ways in which it is different. Our analyses demonstrate the plasticity of the FtsZ protofilament, which may be important for the formation of Z-rings in various sizes for completing cell division."

2.1. Results

9) The authors say: "...the binding did not change significantly in response to any of the nucleotides/nucleotide-analogues tested, and the binding of Mb(S1) to truncated KpFtsZ was also confirmed by gel filtration chromatography, suggesting that the Mb binds to the GTPase domain of FtsZ.". The fact that the binding did not change in response to the nucleotides/nucleotide-analogues does not suggest that the Mb binds to the GTPase domain. Rather, it seems to me to suggest the exact opposite. Indeed, the Mb binds far from FtsZ's GTP binding site.

10) "The binding region of the Mb was distant from the GTP-binding site of FtsZ, which is consistent with the finding that the GTPase activity of KpFtsZ was not affected by Mb binding.". It is likely that the Mb does not affect the formation of the endogenic Z-ring but inhibits its constriction.

11) Could it not also exclude binding partners, such as FtsA?

12) "[The] C2 symmetry"

13) "C-terminus region" should be either "C-terminal region" or "C terminus".

14) "we selected [against] EcFtsZ using two combinatorial phage display libraries"

15) The authors explain their abbreviation in a strange way, saying: "for brevity, hereafter, we will use an abbreviated name for the monobody where the "Ec/KpFtsZ_" segment is omitted". Perhaps be just a bit more explicit: "for brevity, hereafter, we will abbreviate Mb(Ec/KpFtsZS1) to Mb(S1)"

16) When describing the affinity of their monobody for FtsZ, the authors describe the Kd value as "in the [single] μ M range".

2.2. Discussion

17) "its non-impeditive molecular size (~ 10 kDa)". What is "it"? FtsZ or the Mb? Presumably the Mb.

18) I am interested in the axis along which the authors find C2 symmetry. My guess is that it is the filament axis, but I don't see it as given that paired protofilaments of the kind we see here should necessarily have C2 symmetry along this axis. They may very well, but only by coincidence. In any case, the high resolution of the map does support the idea that whatever symmetry the authors have applied is appropriate.

19) I do not get what the authors mean by "the unrestricted binding property of FtsZ to multiple nucleotide-bound states".

20) The authors say: "From our observations, the addition of GTP and GMPCPP dramatically changed the KpFtsZ filament properties to those of single straight filaments, as observed frequently, but GDP and GMPPNP did not. This means that the shapes of the KpFtsZ filaments are not affected by the presence of γ -phosphate (GTP-like or GDP-like)." I don't get this. Adding GTP (with a γ -phosphate) did affect filament morphology. Adding GDP (with no γ -phosphate) did not. Surely what the authors observe is exactly evidence for the established wisdom that the shape of

the filaments is affected by whether or not the bound nucleotide has a γ -phosphate group. This must be made clearer as the author's own data are very obvious on this.

21) I also do not understand how there can be GDP in the cryo-EM structure. According to the authors, the sample contained only FtsZ, Mb, and GMPPNP. As I understand, GMPPNP is not hydrolysable. Was the GMPPNP perhaps contaminated with GDP?

22) Also, while the authors explain the slowly-hydrolysable GMPCPP, they do not explain the non-hydrolysable GMPPNP. In my opinion, both GMPCPP and GMPPNP need a short introduction when first mentioned, in which their use can be justified.

23) The authors say: "As GMPCPP is known as a slowly hydrolyzable analog, GTP hydrolysis would be required for the formation of long and straight FtsZ protofilaments.". However, the observation that - in the presence of GTP and GMPCPP - FtsZ adopts straight filaments surely supports the hypothesis that hydrolysis of FtsZ's bound GTP is necessary for protofilaments to adopt the (curved) conformation seen in helical tubes.

24) In Observation of various types of KpFtsZ filaments, the authors say that they "excluded the GTP and GMPCPP conditions from the specification preparation for cryoEM structure analysis". Unless I am mistaken, they also excluded GDP from their cryo-EM sample preparation. Does my misunderstanding relate to the presence of GDP and not GMPPNP in their cryo-EM map? As mentioned at the beginning, the GTP and GMPCPP filaments would perhaps been more interesting ...

2.3. Methods

25) In the Online Methods, in Effect of nucleotides on monobody binding, the authors have "monobody binding took plate". They surely intend "monobody binding took place".

26) In Negative Staining, the authors have "One side of amorphous carbon grids were hydrophilized". An alternative sentence might begin "Amorphous carbon grids were hydrophilized on one side". Throughout the rest of this paragraph, it would make more sense to say "each grid" than "the grid".

27) In CryoEM specimen preparation and data collection, the authors have "Another KpFtsZ alone sample without adding PC190723 and Mb was also prepared in the same way". I suggest "A KpFtsZ-only sample (without the addition of PC190723 and Mb) was prepared in the same way".

28) The authors have "The grids were blotted ... in a Vitrobot Mark IV chamber (Thermo Fisher Scientific, USA) equilibrated at 4 °C and 100% humidity and then immediately plunged into liquid ethane.". It is surely not the Vitrobot that is plunged into liquid ethane, I encourage a refactor of the sentence.

29) "excess" rather than "excessive" ethane was removed.

30) In CryoEM image processing and model building, I find strange the authors' attribution of their helical parameters as "sub-optimal". Perhaps they intend to say that these estimated parameters were "preliminary" or "approximate".

31) In the caption of Fig 4 | CryoEM analysis of KpFtsZ-Mb double helical tube, the authors remark on panels c and d with "Part of selected 2D class averages of (c) FtsZ-Mb and (d) FtsZ datasets aligned in the descending order of particle numbers from left to right and top to bottom". The expression "in the descending order [...] bottom", while presumably applicable to the CryoSPARC graphic from which these snaps were taken, is not relevant here. Particle numbers can be read off the figure itself. "each FtsZ-Mb complex is shown in [different colors a different color]".

Reviewer #3 (Remarks to the Author):

Text [page 4]: S1 also omitted from the name.

Text [page 5]: The binding strength of the monobody seems rather weak. Would it possible to do better with less target during the affinity selection steps?

Text [page 17]: Please show all the sequences of all five clones in the supplementary information.

Text [page 27]: Why show loop library positions when only monobody is from the side library?

Text [page 27]: Colors should differ from panel a.

Text [page 27]: Did the FG loop length vary in the library design?

Text [page 27]: This is not the typical way to measure Kd. A good estimation though.

Text [page 28]: In addition to the fluorescent images it would be nice to have quantitative data showing increased cell length in cells expressing the monobody.

Text [page 29]: From this structural cartoon, it looks like the monobody is interacting with Fts through its FG loop.

Text [page 32]: How does the monobody interact with two filaments at one time? FG loop on one end, but what residues on the other side are interacting?

Text [page 37]: An unusual y-axis legend. Why Δ ?

Text [page 37]: Background subtracted for the figure?

Text [page 37]: Why not show concentration binding curve with virions?

Text [page 45]: It's hard to see what's going in these overlaid cartoons.

REVIEWER COMMENTS

Reviewer #1 (Remarks to the Author):

The authors of this study have generated a monobody (Mb) that binds to bacterial cell division protein FtsZ. They employed this Mb as a crystallization and assembly chaperone for FtsZ structural studies.

Crystal structures of KpFtsZ-Mb and EcFtsZ-Mb complexes, in two space groups each, show that the Mb binds on a large lateral cavity formed by the central helix H7, the H6-H7 loop, and a beta-sheet in the C-terminal domain. FtsZ in these complexes is in the relaxed (R) conformation, as opposed to the tense (T) conformation active for canonical protofilament assembly. This is possibly the first ligand demonstrated to bind to this FtsZ cavity, which suggests to me the possibility of targeting FtsZ with inhibitors binding to this cavity. It also raises the question of whether natural ligands or partner proteins may bind there.

In solution, the Mb induces KpFtsZ to form 25 nm diameter double helical tubes, made of filaments at an angle of 66.6 degrees with respect to the tube axis. The structure of these FtsZ-Mb tubes has been determined by cryoEM to 2.67 Å resolution. These tubes are possibly related to other FtsZ tubes and mini-rings previously reported in the literature. The geometry of FtsZ-Mb tubes may also be compared to those of curved GDP-tubulin polymers (Wang & Nogales, Nature 2005), tubulin-stathmin complexes (Gigant et al., Cell 2000) and GDP-tubulin rings (Diaz et al., J Mol Biol 1994). All of them are different from microtubules, where the protofilaments align with the microtubule axis.

The work is interesting and technically sound. The manuscript is overall clearly written and figures are informative. Nevertheless, the biological relevance of the results needs to be supported by additional experimental evidence. Detailed comments follow, wishing that some of them may be useful to the authors.

> Thank you for your encouraging comments. We completely agree with you in the point of the possibility of targeting FtsZ with inhibitors, natural ligands, or partner proteins binding to the cavity, so we added the following sentence to the discussion: “The Mb we generated in the present study is the first ligand to bind to the cavity between the central H7 helix, the H6-H7 loop, and the CTD of FtsZ, which highlights the possible binding region for inhibitors, natural ligands, and accessory proteins.”

First, the “microtubule-like” claim for the FtsZ-Mb tubes in the title, abstract and throughout the manuscript is not appropriate, essentially due to distinct filament

organization, as explained above. In addition, the speculation that the FtsZ-Mb tubes 'reflect the structure of the Z-ring in some phases during cell division' (Discussion) is unjustified. Moreover, the authors state that 'a constricted ring should consist of highly curved protofilaments' but a constricted ring could also arise from bundles of short coalescent filaments exhibiting minimal curvature. Bundling of short filaments in clusters, giving rise to a discontinuous Z- ring, is supported by super-resolution microscopy (Bisson-Filho et al., Science 2017; Yang et al., Science 2017; Monteiro et al., Nature 2018).

> Thank you for your suggestion. We removed the word "microtubule-like" and comparison with the microtubules from the title, abstract, and throughout the manuscript. We also deleted the discussion about the relationship between our structure and the Z-ring as you suggested.

Second, the significance of R conformation GDP-FtsZ tubes for bacterial division is uncertain. There is consensus in the field that filaments exist in the T configuration, while the R configuration is mostly present in monomers. Is it possible that the FtsZ-Mb tubes showing a large curvature represent an intermediate of dissociation rather than a configuration in the Z-ring, as proposed by the authors?

> Thank you for your suggestion. We changed the interpretation of the R conformation in the tube structure and redrafted the discussion as you and the reviewer 2 suggested.

Related to the previous comment, would it be possible to study T-conformation polymers, for example by adding the Mb to the FtsZ+GTP sample shown in Figure 3e? Could the authors please clarify which polymers are formed by KpFtsZ with Mb and GTP or GMPCPP? And why the functionally relevant thin polymers (Figure 3e) are excluded from the cryoEM analysis? This would provide more meaningful mechanistic insights. Besides, why add GMPPNP for cryoEM if the bound nucleotide remains GDP from the purification stage (end of page 8)? And why is PC190723, which is known to stabilize the T-conformation filaments, added to the sample used for cryo-EM studies?

> We finally determined the cryoEM structure of a single filament of KpFtsZ in the T conformation at 3.0 Å resolution and found the different binding conformation of GMPCPP compared to the previous crystal structures bound GTP analogues. We added GMPPNP and PC190723 to the double helical tube sample to stabilize the entire structure, because we expected the T conformation in the tubes before solving the structure. Actually, the addition of Mb is enough to stabilize the double helical tube of KpFtsZ.

Third, the results shown in Figure 1g are very interesting, as they indicate that a Z-ring could be formed in the presence of the Mb. Nevertheless, a single image may not be representative and quantification of cell length is required. Importantly, in order to confirm that the Mb-EGFP dots are really Z-rings, the authors should show that FtsZ co-localizes with Mb-EGFP dots, for example by labeling FtsZ with RFP/mCherry. Moreover, Figure 1f shows that the GTPase activity is maintained in the presence of the Mb. How do the authors interpret that GTPase activity is not affected while cell division is inhibited?

> We investigated the cell length distributions of *E. coli* overproducing EGFP and Mb-EGFP and confirmed the obvious difference of the distributions. The results were inserted in Fig 1h. We also observed the cells expressing EcFtsZ-mCherry and Mb-EGFP and confirmed that FtsZ co-localizes with Mb-EGFP dots. The images were inserted in Supplementary Fig. 3. As we described below (about extent of FtsZ polymerization in the GTPase assays), we think that huge 3D structure like bundled filaments is not formed in the GTPase assay, probably due to the absence of supporting layers. From our structure, the binding of Mb does not interfere with the formation of the dimer interface, which is required for GTPase activity. However, the binding of Mb affects the interaction between FtsZ protofilaments, and Z-ring could be stabilized (not sure in the same way as our tube structure) and could not be constricted by Mb binding *in vivo*.

Minor comments:

- Page 2, line 4. The authors claim that high resolution structures are lacking. Of what exactly? Atomic-resolution structures of FtsZ filaments in the T configuration and pseudofilaments in the R conformation have been reported.

> Although many high-resolution crystal structures have been reported, they cannot escape the effects of undesirable crystal packing. That is why high-resolution solution structure is required. To emphasize this point, we modified the sentence as the reviewer 2 suggested in 6): “As the molecular packing that gives rise to crystal structures can have confounding effects on the molecular interactions, ...”

- Page 3, line 6. The text ‘...FtsA and ZipA, ..., and C-terminal peptide of FtsZ and stabilize highly curved FtsZ filaments’ should read ‘...stabilize highly curved FtsZ filaments *in vitro*’.

> We modified the expression as you suggested.

- Page 3, line 8. The ‘treadmilling’ concept deserves a thorough description for a general readership.

> We redrafted the introduction and added the explanation of treadmilling to the sentence as follows, “FtsZ treadmilling, a motion in which the protofilament polymerizes in one end (plus-end) and depolymerizes in the other end (minus-end), drives cell wall synthesis that constricts the Z-ring to form a septum of the cell.”

- Page 4, line 18. It may be useful for the readers to explain better what are ‘the loop library and the side library’.

> We have added the following explanation for it in accordance with the reviewer’s comment: “The loop library diversifies residues in three surface loops (BC-, DE-, and FG-loops) at one end of the FN3 scaffold and the side library diversifies residues in two β -strands (C- and D-strands) in addition to the CD- and FG-loops.”

- Page 5, second paragraph and Figure 1f. What is the extent of FtsZ polymerization in the GTPase assays?

> We measured an absorbance at 600 nm as well as 340 nm in the GTPase activity assay, but no increase of the absorbance at 600 nm was observed. The solution was also not crowded. Therefore, we think that huge 3D structure like the bundled filaments or the double helical tube should not be formed in the condition of the assay, although some thin protofilaments could be generated.

- Page 6, line 16 and Figure 1b are not coherent. In the text ‘C-terminal domain’ is used for the GTPase-activating domain and not for the CTD shown in the figure.

> We replaced CTD in Fig. 1b with CTP (C-terminal peptide) as we call in Introduction.

- Page 6, line 21 and Ext. Figure 5. How do these filaments compare with the reported crystal structure of KpFtsZ lacking Mb (Yoshizawa et al., Acta Cryst D 2020)?

> We found the filament of KpFtsZ–Mb ($P2_12_12_1$) was slightly curved compared to that of KpFtsZ alone (PDB code: 6LL5, Yoshizawa *et al.*, Acta Cryst D 2020). We added the filament of KpFtsZ alone to Supplementary Fig. 5, and modified the sentence as follows, “The molecular architecture of the protofilaments was very similar to that of KpFtsZ (PDB code:6LL5), although the orientation of the molecule is slightly tilted with each other.”

- Page 7, line 14. Cite previous studies.

> We added references.

- Page 9, line 12. 'Non-impeditive' of what? This whole sentence is difficult to understand.

> We completely redrafted the discussion and modified the sentence as follows, "Moreover, this demonstrates the usefulness of Mb, which bears the binding affinity to multiple nucleotide states of FtsZ and non-impeditive molecular size (~10 kDa) as a structure stabilizer to help determine its high-resolution structures by cryoEM as well as crystallography."

- Page 9, line 19. In Figure 6a, how is the -Mb straight model generated? In particular, how is the lateral blue-green interaction deduced? Is it supported by crystal contacts in the structures described in this or past reports?

> That model was inspired by the 2D class averages of KpFtsZ alone (Fig. 4b, lower panel), where the two protofilaments seems to interact each other. Anyway, we removed that model and comparison with the microtubules as the reviewers suggested.

- Page 10, second paragraph. I do not understand the meaning of the sentence 'the shapes of the KpFtsZ filaments are not affected by the presence of gamma phosphate (GTP like or GDP like)'. I also do not understand the sentence 'GTP hydrolysis would be required for the formation of long and straight FtsZ protofilaments'.

> We deleted these sentences and completely redrafted the discussion. Discussion about the relationship between the morphology of FtsZ filaments and the nucleotide condition is described in the fifth paragraph.

- Page 16, line 6. Please include a brief description of the purification method, including buffer compositions and nucleotides/additives used along the purification.

> We added the detailed purification protocols in the paragraph.

- Page 17, second last line. 'took place' instead of 'took plate'.

> We corrected.

- Page 19, line 5. In the crystal structures of FtsZ-Mb a 1:1 FtsZ:Mb complex is observed. For protein crystallization FtsZ 'was mixed with a 1.2x molar excess of Mb'. However, 'final concentrations of FtsZ and Mb were 8 mg/ml and 0.4 mg/ml', which if I am not wrong is a 0.2 molar ratio of Mb to FtsZ. Is this an error?

> We corrected the errors as follows, "Purified KpFtsZtr and EcFtsZtr at a concentration

of 19 mg ml⁻¹ was mixed with 1.2× molar excess of Mb and PC190723 (a known FtsZ inhibitor). The final concentration of KpFtsZtr and EcFtsZtr, and Mb are 11.8 mg ml⁻¹ and 4.0 mg ml⁻¹, respectively.” We also added PC190723 during crystallization for inducing the T conformation, but eventually not bound to the R conformation.

- Table 1. Please report Ramachandran statistics.

> We added Ramachandran statistics to Supplementary Table 1.

Reviewer #2 (Remarks to the Author):

Review of NCOMMS-22-41370-T “High-resolution structure of a microtubule-like tube composed of FtsZ-monobody complexes”

1. Summary

In this article, the authors describe their investigation into the structure of an artificial filament containing the bacterial cytoskeletal protein FtsZ. This filament owes its structure to a 10 kDa monobody that the authors raised against FtsZ, which binds away from FtsZ’s GTPase region. This monobody does not inhibit Z-ring formation (or more precise FtsZ localization) in live cells, but does inhibit division. The authors solved several crystal structures and a cryo-EM structure. They identify various interfacial interactions in their maps, whose effects on the higher-order structure of FtsZ protofilaments they discuss. Despite the authors’ attempts, they did not obtain a cryo-EM structure of the filament without their monobody and they discarded the GTP/GMPCPP-bound forms of FtsZ, which are the most likely *in vivo* form of the filaments, because they were too thin for analysis. To me it is not clear how the monobody-bound filament relates to the biological reality, if at

all. The authors cite FtsZ’s role *in vivo*, which requires that it adopt a range of curvatures. Of course, the monobody is absent in unmodified cells, so this study cannot answer the question of what drives FtsZ filaments to curve during cell division.

> We finally determined the cryoEM structure of a single filament of KpFtsZ (without the monobody) at 3.0 Å resolution and found the different binding conformation of GMPCPP compared to the previous crystal structures bound GTP analogues. This new conformation of GMPCPP may be related to an activated intermediate state of GTP hydrolysis, because the continuous density was observed between the phosphates of

GMPCPP and the T7 loop of the upper molecule.

In the discussion (and the title), the authors make a comparison of their monobody-bound filaments to microtubules. Microtubules are composed of eukaryotic tubulins, to which prokaryotic FtsZ bears homology. However, as the authors point out, their tubular complex differs from microtubules in several significant ways. The direction of the protofilaments is entirely different. The authors' complex forms a 2-start helix, while microtubule subunits form a curved sheet. Moreover, the authors' structure is under the considerable influence of their monobody, which occurs in equal number to FtsZ monomers. In my opinion, the comparison to MTs is unwarranted and misleading and demonstrates the author's inability or unwillingness to interpret their work in the light of biological reality.

> As you suggested, we removed the comparison with the microtubules.

Furthermore, in their brief discussion of the tubular complex's biological relevance, the authors propose that molecules that bind FtsZ in the same manner as their monobody, could induce - according to their size - different inter-protofilament spacings during Z-ring constriction. But during constriction, the curvature of the ring will increase in a continuous fashion. Each different hypothetical crosslinker would presumably rigidly induce a particular spacing. That Z-ring constriction would be quantised like this seems highly unlikely. In fact, the necessary flexibility of the Z-ring surely supports the hypothesis that it is not under the influence of a rigidly crosslinking binding partner.

> As you and the reviewer 1 pointed out, we changed the interpretation of the double helical tube structure and completely redrafted the discussion. Now we consider that the highly curved protofilaments consisting of the R conformation in the tube structure may reflect an intermediate structure of dissociation after GTP hydrolysis, not the partial structure of the Z-ring.

I must conclude that the author's discussion is not insightful or helpful.

While the authors' results are technically impressive, the relevance of their work to FtsZ's role in vivo is at best unclear. I suggest the work needs a completely redrafted title and discussion, and significant editorial adjustments throughout.

2. Individual points

The following points are mostly about specific uses of language, some of which the authors might find helpful.

Introduction

1) “various studies have revealed its molecular mechanisms” Perhaps “molecular structure” would be a better choice of words. To my mind, a mechanism is a process through which some action is performed. It is also not clear to me to whether “it” refers to FtsZ or the Z-ring.

> We modified the expression as follows, “various studies have revealed the molecular structures of FtsZ”.

2) The last part of the first sentence is poorly worded: “plays a key role in a complex called divisome that functions during cell division”. The divisome doesn’t just function during division - it carries out division.

> We modified the expression as follows, “plays a key role in a complex called divisome, which carries out cell division”.

3) The authors write: “FtsZ polymerizes into protofilaments and a ring-shaped structure (Z-ring) with GTP, which is located in the middle of the cell and constricts to provide force for dividing the cell in half.” It is the Z-ring, not GTP, that is located in the middle of the cell. To clear this, up might I recommend: “With GTP, FtsZ polymerizes into protofilaments and a ring-shaped structure (Z-ring), which is located in the middle of the cell and constricts to provide force for dividing the cell in half.”

> We redrafted the introduction and modified the sentence as follows, “With GTP, FtsZ polymerizes into protofilaments and a ring-shaped structure (Z-ring), which is located in the middle of the cell and recruits more than 30 partner proteins.”

4) FtsZ is tethered to the cell membrane [through by] other proteins ... This first paragraph seems to contradict itself, saying initially that it is Z-ring constriction that provides the force necessary to divide the cell, and then that it is the remodelling of the cell wall. If the authors wish to communicate that it is not known from where the necessary force comes, then they should say so.

> We redrafted the introduction.

5) Rather than “the molecular mechanism leading to the FtsZ function”, I suggest maybe

“the molecular mechanism through which FtsZ [polymerises]”.

> We modified the expression as follows, “the molecular mechanism through which FtsZ polymerizes and depolymerizes”.

6) The authors have: “these crystal structures cannot escape the effect of crystal packing”. Crystal structures in general cannot escape the effects of crystal packing. This sentence misses the point that these effects are often undesirable. I suggest: “The crystal packing that gives rise to crystal structures can have confounding effects on molecular interactions.”

> We modified the sentence as follows, “As the molecular packing that gives rise to crystal structures can have confounding effects on the molecular interactions, ...”.

7) Rather than “inhibits proper cell division to elongate the cells”, how about “inhibits proper cell division, causing the cells to elongate”:

> We modified the sentence as follows, “inhibited proper cell division, causing the cells to elongate.”

8) “Our structural analyses showed the structural similarity and difference between the microtubule and FtsZ tube and the plasticity of the FtsZ protofilament, which may be important for the formation of Z-rings in various sizes for completing cell division.”. I'm not certain what this sentence is intended to communicate. Also, the word “structural” is not necessary. By now, it is well established that we are talking about the structure of FtsZ filaments. Perhaps the authors would be better served by dividing this sentence. “We see ways in which the FtsZ tube is similar to microtubules and ways in which it is different. Our analyses demonstrate the plasticity of the FtsZ protofilament, which may be important for the formation of Z-rings in various sizes for completing cell division.”

> We deleted the first sentence because we removed the comparison with the microtubules. We modified the second sentence as follows, “Our present study highlights the common and distinct structural features of FtsZ between those in crystal and solution as well as the usefulness of Mb as a structural stabilizer and the physiological roles of the T and R conformations in treadmilling and divisome formation.”

2.1. Results

9) The authors say: “...the binding did not change significantly in response to any of the

nucleotides/nucleotide-analogues tested, and the binding of Mb(S1) to truncated KpFtsZ was also confirmed by gel filtration chromatography, suggesting that the Mb binds to the GTPase domain of FtsZ.”. The fact that the binding did not change in response to the nucleotides/nucleotide-analogues does not suggest that the Mb binds to the GTPase domain. Rather, it seems to me to suggest the exact opposite. Indeed, the Mb binds far from FtsZ’s GTP binding site.

> We modified the sentence as follows, “The results showed that the binding did not change significantly in response to any of the nucleotides/nucleotide-analogs tested, and the Mb binding to truncated KpFtsZ (residues 11–316; KpFtsZtr) was also confirmed by gel filtration chromatography, suggesting that the Mb binding site is located in the GTPase domain but not overlapped with the GTP binding site.”

10) “The binding region of the Mb was distant from the GTP-binding site of FtsZ, which is consistent with the finding that the GTPase activity of KpFtsZ was not affected by Mb binding.”. It is likely that the Mb does not affect the formation of the endogenic Z-ring but inhibits its constriction.

> As we described in the response to the reviewer 1, we think that huge 3D structure like bundled filaments is not formed in the GTPase assay, probably due to the absence of supporting layers. From our structure, the binding of Mb does not interfere with the formation of the dimer interface, which is required for GTPase activity. However, the binding of Mb affects the interaction between FtsZ protofilaments, and Z-ring could be stabilized (not sure in the same way as our tube structure) and could not be constricted by Mb binding *in vivo*.

11) Could it not also exclude binding partners, such as FtsA?

> We think that Mb and binding partners like FtsA rarely competes, because most of the binding partners recognize CTP, which is far from the GTPase domain of FtsZ.

12) “[The] C2 symmetry”

> We deleted “The”.

13) “C-terminus region” should be either “C-terminal region” or “C terminus”.

> We changed the expression from “C-terminus region” to “C-terminal region”.

14) “we selected [against] EcFtsZ using two combinatorial phage display libraries”

> We added the word “against”.

15) The authors explain their abbreviation in a strange way, saying: “for brevity, hereafter, we will use an abbreviated name for the monobody where the “Ec/KpFtsZ_” segment is omitted”. Perhaps be just a bit more explicit: “for brevity, hereafter, we will abbreviate Mb(Ec/KpFtsZS1) to Mb(S1)”

> We modified the sentence as follows, “In these reads, all isolates had absolutely identical sequence derived from the side library, and we named it Mb(Ec/KpFtsZ_S1), hereafter abbreviated as Mb for brevity.”

16) When describing the affinity of their monobody for FtsZ, the authors describe the K_d value as “in the [single] μM range”.

> We modified the expression as “in a low μM range”, because we also measured the K_d value of 24 and 20 μM using SPR as the reviewer 3 suggested.

2.2. Discussion

17) “its non-impeditive molecular size (~10 kDa)”. What is “it”? FtsZ or the Mb? Presumably the Mb.

> We completely redrafted the discussion and modified the sentence as follows, “Moreover, this demonstrates the usefulness of Mb, which bears the binding affinity to multiple nucleotide states of FtsZ and non-impeditive molecular size (~10 kDa) as a structure stabilizer to help determine its high-resolution structures by cryoEM as well as crystallography.”

18) I am interested in the axis along which the authors find C2 symmetry. My guess is that it is the filament axis, but I don’t see it as given that paired protofilaments of the kind we see here should necessarily have C2 symmetry along this axis. They may very well, but only by coincidence. In any case, the high resolution of the map does support the idea that whatever symmetry the authors have applied is appropriate.

> We realized C2 symmetry along the filament axis when we ran the map symmetry job in PHENIX. Then we re-ran the helix refine job with C2 symmetry and found the map resolution improved. We agree that C2 symmetry is not necessarily required for this kind of paired filament. We think the binding of Mb helps to reduce structural flexibility and reconstruct highly periodic structure.

19) I do not get what the authors mean by “the unrestricted binding property of FtsZ to

multiple nucleotide-bound states”.

> As described in 17), we modified the sentence as follows, “Moreover, this demonstrates the usefulness of Mb, which bears the binding affinity to multiple nucleotide states of FtsZ and non-impeditive molecular size (~10 kDa) as a structure stabilizer to help determine its high-resolution structures by cryoEM as well as crystallography.”

20) The authors say: “From our observations, the addition of GTP and GMPCPP dramatically changed the KpFtsZ filament properties to those of single straight filaments, as observed frequently, but GDP and GMPPNP did not. This means that the shapes of the KpFtsZ filaments are not affected by the presence of γ -phosphate (GTP-like or GDP-like).” I don’t get this. Adding GTP (with a γ -phosphate) did affect filament morphology. Adding GDP (with no γ -phosphate) did not. Surely what the authors observe is exactly evidence for the established wisdom that the shape of the filaments is affected by whether or not the bound nucleotide has a γ -phosphate group. This must be made clearer as the author’s own data are very obvious on this.

> We deleted these sentences and completely redrafted the discussion. Discussion about the relationship between the morphology of FtsZ filaments and the nucleotide condition is described in the fifth paragraph.

21) I also do not understand how there can be GDP in the cryo-EM structure. According to the authors, the sample contained only FtsZ, Mb, and GMPPNP. As I understand, GMPPNP is not hydrolysable. Was the GMPPNP perhaps contaminated with GDP?

> FtsZ always binds GDP even if we did not add it during purification. Our previous SaFtsZ and KpFtsZ crystal structures also contain GDP, which is not supplemented with (J. Fujita *et al.*, *J. Struct. Biol.* 198, 65 (2017) and T. Yoshizawa *et al.*, *Acta Cryst. F* 76, 86 (2020)). GDP should be brought from *E. coli* in the cultivation step. To clarify this point, we modified the sentence in the result as follows, “We found GDP in the binding pocket instead of GMPPNP, indicating that added GMPPNP did not replace GDP that was kept bound in KpFtsZ during the purification from *E. coli* cells.” We also added the following sentence in the fifth paragraph of discussion: “Notably, FtsZ always binds endogenous GDP brought in from *E. coli* cells even without adding nucleotides, as we have experienced in our previous crystal structure analyses.”

22) Also, while the authors explain the slowly-hydrolysable GMPCPP, they do not explain the non-hydrolysable GMPPNP. In my opinion, both GMPCPP and GMPPNP need a short introduction when first mentioned, in which their use can be justified.

> We added short introductions of GMPPNP and GMPCPP when first mentioned.

23) The authors say: “As GMPCPP is known as a slowly hydrolyzable analog, GTP hydrolysis would be required for the formation of long and straight FtsZ protofilaments.”. However, the observation that - in the presence of GTP and GMPCPP - FtsZ adopts straight filaments surely supports the hypothesis that hydrolysis of FtsZ's bound GTP is necessary for protofilaments to adopt the (curved) conformation seen in helical tubes.

> We deleted these sentences and completely redrafted the discussion. Discussion about the relationship between the morphology of FtsZ filaments and the nucleotide condition is described in the fifth paragraph.

24) In Observation of various types of KpFtsZ filaments, the authors say that they “excluded the GTP and GMPCPP conditions from the specification preparation for cryoEM structure analysis”. Unless I am mistaken, they also excluded GDP from their cryo-EM sample preparation. Does my misunderstanding relate to the presence of GDP and not GMPPNP in their cryo-EM map? As mentioned at the beginning, the GTP and GMPCPP filaments would perhaps been more interesting ...

> As we described in 21), FtsZ always binds GDP, and therefore we consider that there is no difference between the samples of FtsZ alone and FtsZ supplemented with GDP. The absence of GMPPNP in our cryoEM tube structure indicates that GMPPNP could not be replaced with GDP because of its low affinity (Scheffers, *et al.*, *Mol. Microbiol.* 35, 1211 (2000)), as we added to the fifth paragraphs of the discussion. We finally determined the cryoEM structure of a single filament of KpFtsZ (without the monobody) at 3.0 Å resolution.

2.3. Methods

25) In the Online Methods, in Effect of nucleotides on monobody binding, the authors have “monobody binding took plate”. They surely intend “monobody binding took place”.

> We corrected.

26) In Negative Staining, the authors have “One side of amorphous carbon grids were hydrophilized”. An alternative sentence might begin “Amorphous carbon grids were hydrophilized on one side”. Throughout the rest of this paragraph, it would make more sense to say “each grid” than “the grid”.

> We modified the expressions as you suggested.

27) In CryoEM specimen preparation and data collection, the authors have “Another KpFtsZ alone sample without adding PC190723 and Mb was also prepared in the same way”. I suggest “A KpFtsZ-only sample (without the addition of PC190723 and Mb) was prepared in the same way”.

> We modified the sentence as you suggested.

28) The authors have “The grids were blotted ... in a Vitrobot Mark IV chamber (Thermo Fisher Scientific, USA) equilibrated at 4 °C and 100% humidity and then immediately plunged into liquid ethane.”. It is surely not the Vitrobot that is plunged into liquid ethane, I encourage a refactor of the sentence.

> We modified the sentences as follows, “3 µl of the solution was applied to the glow-discharged grids in a Vitrobot Mark IV chamber (Thermo Fisher Scientific, USA) equilibrated at 4 °C and 100% humidity. The grids were blotted with a force of 0 and a time of 3 sec (KpFtsZ–Mb double helical tube) or with a force of –10 and a time of 1.5 sec (KpFtsZ double helical tube and single filament) and then immediately plunged into liquid ethane.”

29) “excess” rather than “excessive” ethane was removed.

> We modified the expression as you suggested.

30) In CryoEM image processing and model building, I find strange the authors’ attribution of their helical parameters as “sub-optimal”. Perhaps they intend to say that these estimated parameters were “preliminary” or “approximate”.

> We changed the word from “sub-optimal” to “approximate”.

31) In the caption of Fig 4 | CryoEM analysis of KpFtsZ-Mb double helical tube, the authors remark on panels c and d with “Part of selected 2D class averages of (c) FtsZ-Mb and (d) FtsZ datasets aligned in the descending order of particle numbers from left to right and top to bottom”. The expression “in the descending order [...] bottom”, while presumably applicable to the CryoSPARC graphic from which these snaps were taken, is not relevant here. Particle numbers can be read off the figure itself. “each FtsZ-Mb complex is shown in [different colors a different color]”.

> We deleted the expression “aligned in the descending order of particle numbers from left to right and top to bottom”, and changed the expression from “different colors” to “a

different color”.

Reviewer #3 (Remarks to the Author):

Text [page 4]: S1 also omitted from the name.

> We deleted.

Text [page 5]: The binding strength of the monobody seems rather weak. Would it possible to do better with less target during the affinity selection steps?

> We would say yes to the question. The way the reviewer suggested as well as affinity maturation strategy using a yeast surface display system in which we previously established (Tanaka, S.-i., *et al.*, *Nature Chem. Biol.* 2015 and Tanaka, S.-i., *et al.*, *ACS Chem. Biol.* 2018) would work for generating stronger binders. However, in this work, we have been mainly focusing on using monobody for crystallographic and cryoEM analyses, where the protein concentration used is often in ~hundreds μM range. Because we expected that such high protein concentration could compensate for low affinity and indeed the stable complex formation of FtsZ-Mb in concentrations of $\sim 600 \mu\text{M}$ was obvious when analyzed by SEC, we decided to go with Mb(Ec/KpFtsZ_S1) though the affinity is not high.

Text [page 17]: Please show all the sequences of all five clones in the supplementary information.

> We apologize for unclear and imprecise descriptions. We picked five isolates exhibiting specific binding to KpFtsZ and EcFtsZ and subjected them to DNA sequencing. In these reads, we realized that all five isolates had absolutely identical sequence. We have revised the text to precisely describe the process.

Text [page 27]: Why show loop library positions when only monobody is from the side library?

> We agree with the reviewer. We have revised panel (a) in Fig. 1 to show only side library positions. Because we used the loop library in addition to the side library for screening campaign, loop library positions are also provided in Supplementary Fig. 1.

Text [page 27]: Colors should differ from panel a.

> We thank the reviewer for pointing out this. We have revised panel (b) in Fig. 1 with changing letters' color to black.

Text [page 27]: Did the FG loop length vary in the library design?

> Yes, we designed the library based on the ref. 7 in the online Methods (Koide, A. *et al.*, *J. Mol. Biol.* 415, 393 (2012)).

Text [page 27]: This is not the typical way to measure Kd. A good estimation though.

> We added the data of SPR measurement in Fig. 1d and moved the data of yeast surface display to Supplementary Fig. 2b. We also revised the text to describe both of the methods.

Text [page 28]: In addition to the fluorescent images it would be nice to have quantitative data showing increased cell length in cells expressing the monobody.

> We investigated the cell length distributions of *E. coli* overproducing EGFP and Mb-EGFP and confirmed the obvious difference of the distributions. The results were inserted in Fig 1h. We also observed the cells expressing EcFtsZ-mCherry and Mb-EGFP and confirmed that FtsZ co-localizes with Mb-EGFP dots. The images were inserted in Supplementary Fig. 3.

Text [page 29]: From this structural cartoon, it looks like the monobody is interacting with Fts though its FG loop.

> We think the interaction through the FG loop is quite possible, because the residues in the FG loop are also diversified in the side library. To clarify this point, we added the following sentence to the first paragraph of results, "The loop library diversifies residues in three surface loops (BC-, DE-, and FG-loops) at one end of the FN3 scaffold and the side library diversifies residues in two β -strands (C- and D-strands) in addition to the CD- and FG-loops."

Text [page 32]: How does the monobody interact with two filaments at one time? FG loop on one end, but what residues on the other side are interacting?

> As shown in Fig. 4h and 4i, FG loop of Mb interacts with FtsZ, which is also confirmed in the crystal structures, and the opposite side of Mb interacts with FtsZ in the other protofilament.

Text [page 37]: An unusual y-axis legend. Why Δ ?

> We thank the reviewer for pointing out this. We have deleted Δ from the y-axis legend

of Supplementary Fig. 2a.

Text [page 37]: Background subtracted for the figure?

> We have updated the figure. Please see Supplementary Fig. 2a.

Text [page 37]: Why not show concentration binding curve with virions?

> In response to the reviewer's comment, we have updated the Supplementary Fig. 2a with the results of titered phage ELISA.

Text [page 45]: It's hard to see what's going in these overlaid cartoons.

> We deleted the figure as we removed the comparison with the microtubules.

REVIEWER COMMENTS

Reviewer #1 (Remarks to the Author):

During the first round of revision, my main concern related to the biological significance of the results. In this revised version the authors have removed interpretations on their helical tube structure as a microtubule-like entity or as representative of a configuration in the Z-ring. While this eases rationalization of their KpFtsZ-Mb structures, those results alone would, in my opinion, be more suitable for a specialized journal. Nevertheless, the authors have succeeded in obtaining the cryo-EM structure of a KpFtsZ protofilament in solution at 3 Å resolution, which exhibits a T conformation as expected for the canonical filament state. This result is remarkable and, together with improved co-localization experiments and the FtsZ-Mb structures, constitutes a valuable result for the field of bacterial cell division.

Specific comments:

In the new section termed 'Comparison of FtsZ in the monomeric and protofilament structures', the authors compare their KpFtsZ protofilament structure with that of SaFtsZ bound to GTP- γ S, but this analog is unable to induce filament formation as opposed to GMPCPP in their structure. I suggest they include a comparison with SaFtsZ bound to GDP, beryllium fluoride and magnesium (Reference 27), which mimics the GTP-bound state. While I agree with the authors that mechanistic analysis is risky at 3.0 Å resolution, they claim that the gamma-phosphate in their structure approaches the T7 loop as opposed to the GTP- γ S structure. Comparison with the GTP-bound mimetic could shed light into this observation. Importantly, map representations in Fig 5e-f should be improved to understand how they built the nucleotide in the map. Can the authors indicate in the figure the position of catalytic residues in the T7 loop?

Regarding in vivo fluorescence experiments, could Mb interfere with the switch between the R and the T conformations, which is critical for treadmilling? This would explain that FtsZ and Mb colocalize while cell division is compromised. I would also recommend to improve the legend in Supp Fig 3.

Reviewer #2 (Remarks to the Author):

Re-review NCOMMS-22-41370-T, Fujita et al., "Structures of a single protofilament of FtsZ and a double-helical tube of FtsZ-monobody complex"

The authors raised a monobody against FtsZ, a polymerising protein involved in bacterial cell division that forms the mid-cell Z-ring. In cells, the authors' monobody does not allow cells to divide, though Z-rings do still seem to form. In vitro, where FtsZ is known to form tubes and mini-rings, the monobody helps to mold FtsZ into 25 nm-diameter double-helical tubes, amongst many other structures. The authors present a 2.7 Å cryo-EM structure of these tubes and a gamut of FtsZ-Mb crystal structures.

Since the first submission and review, the authors have introduced many editorial changes to try to deal with the plethora of comments they were presented with. For example, it is good that the comparisons of the FtsZ-Mb tubes to microtubules have been removed everywhere as they were unwarranted.

Since the first round of review, the authors have now also determined a 3.0 Å cryo-EM structure of a straight KpFtsZ protofilament. This filament contains GMPCPP bound in a manner that they believe to represent a hydrolysis intermediate. This structure, at first sight at least, is arguably the most noteworthy work of this manuscript, as they state that it is the first ever near-atomic resolution cryo-EM structure of an FtsZ filament.

Some of the work is technically impressive, but as the other reviewers have also noted, its biological relevance is at best unclear. The discussion remains confused and we get the impression

that concepts describing FtsZ function in cells and during treadmilling have perhaps not been fully grasped by the authors, most notably: PMID: 28465423, PMID: 32763138 and PMID: 36989372, work that is directly relevant to the structures that the authors have obtained and try to put into a biological or at least mechanistic context.

In summary, two major points remain: the manuscript is ordered by how the work proceeded, which is unfortunately not the best way to present it. Why are we reading this? What was the question when the authors started to work on this? To our mind, the authors tried to find out what structures KpFtsZ has in its various nucleotide and polymerization states. These findings then needed to be put into the framework of prior knowledge that the closed / R state occurs in monomeric forms, most likely independent of nucleotide state, and that the open / T state occurs in filaments, again most likely independent of nucleotide state. In our opinion, the authors' work supports this model (that is now pretty much accepted by the actin and tubulin fields) and the work could be presented with great clarity if it was followed.

Second, we are not convinced that the new KpFtsZ cryo-EM protofilament structure warrants the label of 3 Å resolution. Supplementary Figure 10 is somewhat alarming since the density really does not describe the nucleotide well. The density for the rest of the protein also looks blobby, although not enough plots are provided at enough detail to be clear about this. More evidence is needed to support the claim of 3 Å resolution, including density close-ups, orientation distributions and anisotropy analysis, as well as feature-based resolution estimation.

We are really sorry, but there remain major issues, some of them are new. It is an important subject but the field has moved on a lot and the work has the potential to be presented with much more impact and clarity.

Some more specific comments:

- 1) Has the KpFtsZ GMPCPP structure been compared to AlphaFold models?
- 2) As mentioned already, we think the manuscript needs to start with the FtsZ protofilament structure (since least modified) and then the monobody complexes. We note that the new title also suggests this order.
- 3) There are many grammatical mistakes that we did not list in this round because of the major issues identified. Hopefully the authors can get some help with this in the future. It might be a good idea to employ professional editing services or at least AI-based tools that detect problems and suggest improvements such as Grammarly. We are aware that the authors are not native speakers and they should not be overly criticized for these issues.
- 4) Line 63: "... have also revealed the filamentous structures in the R conformation" – this is not correct: the structure was pseudo-filamentous as we commented on as reviewers at the time - and the authors (some the same here) agreed.
- 5) Line 70: we think that the sentence is overstating what is still missing: the crystal structures in the T / closed form are pretty accurate as the authors comment in the present manuscript.
- 6) Supp Figures 9 and 10: We are not convinced that the density is good enough. Especially Figure S10 reveals that the density is blobby and does not describe the nucleotide well at all. Is it possible that the reconstruction suffered from preferred orientation bias? Could you please show the orientation distribution of the particles that went into the final reconstruction? Wagstaff et al reported very similar problems (Wagstaff et al Sci Adv [2023]). Also needed: density close-ups, anisotropy analysis and feature-based resolution estimation.
- 7) Line 266: see last point: we are not convinced that the GMPCPP density in the cryo-EM structure justifies this statement.
- 8) Line 325 and below: we disagree with the idea that the ring forms because FtsZ hydrolyses GTP to GDP and then curves. As the text says, GDP FtsZ

depolymerizes and is not part of the ring. This drives treadmilling instead.

Some editorial comments, in case they help, since it is fully understood that the authors are not native speakers:

9) The title misses an article we think: "Structures of a single protofilament of FtsZ and a double-helical tube of an FtsZ-monobody complex". Maybe better: "Structures of a single protofilament of FtsZ and in complex with a monobody"

10) Introduction: Rather than "In this study, based on the tenth human fibronectin type III domain (FN3), we developed a monobody (Mb), a synthetic protein that binds to both KpFtsZ and EcFtsZ at high affinity, to aid high-resolution structure determination of the FtsZ protofilament", have something like "In order to aid high-resolution structure determination of the FtsZ protofilament, we developed a monobody (Mb) that binds to both KpFtsZ and EcFtsZ with high affinity." There is no need for a detailed explanation of what a monobody is. Also, "with" rather than "at". "at" makes it sound as if the authors are describing the conditions under which binding occurs.

11) Results: "the orientation of the molecule is slightly tilted with each other." What? Maybe "the relative orientation of the molecules is slightly different."?

12) Discussion: May we suggest in the discussion, instead of "The difference in the polymer morphology formed in the presence of two GTP analogs, GMPPNP for the double-helical tube and GMPCPP for the single protofilament, can be explained by the fact that GMPPNP did not replace endogenous GDP as revealed in the structure of the tube with bound GDP despite the addition of GMPPNP in solution (Fig. 4g), which is possibly because of the lower affinity of GMPPNP for FtsZ than GDP. Notably, FtsZ always binds endogenous GDP brought in from E. coli cells even without adding nucleotides, as we have experienced in our previous crystal structure analyses." to have "We have experienced in our previous crystal structure analyses that without added nucleotides, FtsZ binds endogenous GDP from E. coli cells. The structure of our tube reveals that, despite the addition of GMPPNP in solution (Fig. 4g), endogenous GDP was not replaced. This may be because GMPPNP's affinity for FtsZ is lower than GDP's." Different polymer morphologies seem to form depending the nucleotide or nucleotide analog bound: double-helical tubes in the case of GMPPNP and single protofilaments for GDP." The wording as it stands now seems to say that the tube structure is what FtsZ would do in the presence of GMPPNP, when the authors want to say that the tube structure has bound GDP.

13) The authors have used "intimate" a couple of times. This is perhaps a bit too poetic: "close" is just fine.

14) Change "stabilized the tubular structure of KpFtsZ in solution (Fig. 3) for its high-resolution cryoEM study (Fig. 4)." to "stabilized the tubular structure of KpFtsZ in solution (Fig. 3) for its study by high-resolution cryo-EM (Fig. 4)."

15) In the discussion, the authors write "Now that KpFtsZ can be in either of the T and R conformations depending on the bound nucleotide in different polymer structures formed in solution, not only for the intramolecular structure but also for the intermolecular interactions along the protofilament with different curvatures, there is no doubt that the roles and mechanisms of structural transition of FtsZ between the T and R conformations are universally shared by a wide range of species and is canonical." We think that the authors' claims are a bit too bold. They may be right, but what they have shown here does not justify language like "no doubt" and "universally". We would also argue strongly that the structures solved do not support the idea that nucleotide state controls the closed / R to open / T transition, given that monobodies are involved and all but one structure are constraint by crystal contacts (as the authors state themselves). There is no open / T structure that is monomeric, for example, but there are closed / R structures in all nucleotide states and they are all monomeric (or at least not properly polymeric with the T7 loop able to activate hydrolysis).

REVIEWER COMMENTS

Reviewer #1 (Remarks to the Author):

During the first round of revision, my main concern related to the biological significance of the results. In this revised version the authors have removed interpretations on their helical tube structure as a microtubule-like entity or as representative of a configuration in the Z-ring. While this eases rationalization of their KpFtsZ-Mb structures, those results alone would, in my opinion, be more suitable for a specialized journal. Nevertheless, the authors have succeeded in obtaining the cryo-EM structure of a KpFtsZ protofilament in solution at 3 Å resolution, which exhibits a T conformation as expected for the canonical filament state. This result is remarkable and, together with improved co-localization experiments and the FtsZ-Mb structures, constitutes a valuable result for the field of bacterial cell division.

> Thank you for your encouraging comments.

Specific comments:

In the new section termed ‘Comparison of FtsZ in the monomeric and protofilament structures’, the authors compare their KpFtsZ protofilament structure with that of SaFtsZ bound to GTP- γ S, but this analog is unable to induce filament formation as opposed to GMPCPP in their structure. I suggest they include a comparison with SaFtsZ bound to GDP, beryllium fluoride and magnesium (Reference 27), which mimics the GTP-bound state.

> We replaced SaFtsZ–GTP γ S complex with SaFtsZ complexed with GDP, BeF₃⁻, and Mg²⁺ (PDB code: 7OHK) in new Supplementary Fig. 3a and 3b and modified the main text, because the nucleotide conformations are almost identical between the GTP γ S and GDP, BeF₃⁻, Mg²⁺ complexes.

While I agree with the authors that mechanistic analysis is risky at 3.0 Å resolution, they claim that the gamma-phosphate in their structure approaches the T7 loop as opposed to the GTP- γ S structure. Comparison with the GTP-bound mimetic could shed light into this observation. Importantly, map representations in Fig 5e-f should be improved to understand how they built the nucleotide in the map. Can the authors indicate in the figure the position of catalytic residues in the T7 loop?

> We toned down our claim about the mechanism of GTP hydrolysis, because we could not determine the accurate positions and conformations of GMPCPP and the T7 loop, especially the catalytic residues, from our map. As described in the main text, we put GMPCPP based on the strong density that is likely corresponding to the three phosphates (Fig. 2f, higher-level contoured map shown in magenta). We consider that the GMPCPP and the T7 loop are in a mixture of multiple conformations, but unfortunately, we could not separate them by classification.

Regarding in vivo fluorescence experiments, could Mb interfere with the switch between the R and the T conformations, which is critical for treadmilling? This would explain that FtsZ and Mb colocalize while cell division is compromised. I would also recommend to improve the legend in Supp Fig 3.

> We assume that the monomeric KpFtsZ–Mb complex in the R conformation can polymerize and change to the T conformation, because the addition of Mb does not have much effect on the GTPase activity (Fig. 3f). The colocalization of FtsZ and Mb in cells may indicate that Z-rings form, but lateral interactions between FtsZ protofilaments and constriction of the Z-ring can be interfered by Mbs. Therefore, we are not completely sure whether Mb interferes with the switch between the T and the R conformations, but Mb may have some effects on the switch, because the Mb binding site of FtsZ (the H7 helix, the H6-H7 loop, and the CTD) shows large structural changes between the two conformations. We also revised the legend in the new Supplementary Fig. 6.

Reviewer #2 (Remarks to the Author):

Re-review NCOMMS-22-41370-T, Fujita et al., “Structures of a single protofilament of FtsZ and a double-helical tube of FtsZ-monobody complex”

The authors raised a monobody against FtsZ, a polymerising protein involved in bacterial cell division that forms the mid-cell Z-ring. In cells, the authors' monobody does not allow cells to divide, though Z-rings do still seem to form. In vitro, where FtsZ is known to form tubes and mini-rings, the monobody helps to mold FtsZ into 25 nm-diameter double-helical tubes, amongst many other structures. The authors present a 2.7 Å cryo-EM structure of these tubes and a gamut of FtsZ-Mb crystal structures.

Since the first submission and review, the authors have introduced many editorial changes

to try to deal with the plethora of comments they were presented with. For example, it is good that the comparisons of the FtsZ-Mb tubes to microtubules have been removed everywhere as they were unwarranted.

> Thank you for your encouraging comments.

Since the first round of review, the authors have now also determined a 3.0 Å cryo-EM structure of a straight KpFtsZ protofilament. This filament contains GMPCPP bound in a manner that they believe to represent a hydrolysis intermediate. This structure, at first sight at least, is arguably the most noteworthy work of this manuscript, as they state that it is the first ever near-atomic resolution cryo-EM structure of an FtsZ filament.

Some of the work is technically impressive, but as the other reviewers have also noted, its biological relevance is at best unclear. The discussion remains confused and we get the impression that concepts describing FtsZ function in cells and during treadmilling have perhaps not been fully grasped by the authors, most notably: PMID: 28465423, PMID: 32763138 and PMID: 36989372, work that is directly relevant to the structures that the authors have obtained and try to put into a biological or at least mechanistic context.

> As described below, we revised the discussion to put our results into the framework of the cytomotive switch concepts in the papers you suggested.

In summary, two major points remain: the manuscript is ordered by how the work proceeded, which is unfortunately not the best way to present it. Why are we reading this? What was the question when the authors started to work on this? To our mind, the authors tried to find out what structures KpFtsZ has in its various nucleotide and polymerization states. These findings then needed to be put into the framework of prior knowledge that the closed / R state occurs in monomeric forms, most likely independent of nucleotide state, and that the open / T state occurs in filaments, again most likely independent of nucleotide state. In our opinion, the authors' work supports this model (that is now pretty much accepted by the actin and tubulin fields) and the work could be presented with great clarity if it was followed.

> We have changed the order of the results as you suggested. In the introduction, we added the following sentence to spot the question we focused on in this paper, "However, due to the lack of high-resolution solution structures of the FtsZ protofilament, the relationships between these different types of polymers, the monomeric T/R conformations, and the bound nucleotides remain to be elucidated."

Then we added the following sentences to the second paragraph in the discussion, "This

is the first case not only for the solution structure of FtsZ single protofilament at the main chain resolvable resolution but also for the T conformation protofilament in the species whose crystal structure shows the R conformation in the pseudo-filamentous structures. Additionally, all the crystal structures of KpFtsZ-Mb and EcFtsZ-Mb in this study adopt the R conformation regardless of either straight or curved protofilament-like structures. The KpFtsZ protofilament structure ensures that the T conformation is preferable in the polymerized form and the R conformations observed in the pseudo-filamentous structures are stabilized by molecular interactions in the crystals.”

We also added the following sentences to the fourth paragraph in the discussion, “Now we confirm that the solution structure of KpFtsZ single protofilament adopts the T conformation in the presence of GMPCPP, in contrast to the crystal structures. Our results may support the cytomotive switch model, where the conformation switch between the T and R states is induced by the intermolecular interactions during polymerization and depolymerization and not by the changes in nucleotide states. The ambiguity of the cryoEM map corresponding to the GMPCPP and the T7 loop might indicate that they are in a mixture of multiple conformations in the protofilaments with the T conformation. Conversely, the T conformation of FtsZ in a protofilament may not be strongly affected by the nucleotide states and the conformations of the T7 loop as far as FtsZ polymerizes.”

Second, we are not convinced that the new KpFtsZ cryo-EM protofilament structure warrants the label of 3 Å resolution. Supplementary Figure 10 is somewhat alarming since the density really does not describe the nucleotide well. The density for the rest of the protein also looks blobby, although not enough plots are provided at enough detail to be clear about this. More evidence is needed to support the claim of 3 Å resolution, including density close-ups, orientation distributions and anisotropy analysis, as well as feature-based resolution estimation.

> We agree that our cryoEM map of the single protofilament lacks the quality of a normal 3 Å resolution map, probably because of the biased particle orientation distribution and low sphericity derived from the flexible filamentous structure without helical twist, although the β -sheet in the CTD and the H7 helix was resolved well (Fig. 2d). We also tried to improve the map quality by more classification and refinement with different parameters, but could not. Therefore, we deleted our claim about 3 Å resolution and intermediate state of GTP hydrolysis throughout the manuscript. We state that the resolution of the final reconstructed 3D map reached 3.03 Å (FSC = 0.143 in cryoSPARC) only in one part of the results, and we also added different estimations such as 3DFSC and d_{model} as well as density close-ups, as described below.

We are really sorry, but there remain major issues, some of them are new. It is an important subject but the field has moved on a lot and the work has the potential to be presented with much more impact and clarity.

Some more specific comments:

1) Has the KpFtsZ GMPCPP structure been compared to AlphaFold models?

> We performed the structure prediction of full-length and truncated KpFtsZ using AlphaFold2. All predicted structures are in the R conformation and similar to our crystal and cryoEM tube structures, probably because they are affected by the published crystal structure of KpFtsZ. Therefore, we considered that the comparison between AlphaFold2 and cryoEM single protofilament structures does not give new information compared to the comparison in this manuscript.

2) As mentioned already, we think the manuscript needs to start with the FtsZ protofilament structure (since least modified) and then the monobody complexes. We note that the new title also suggests this order.

> We rearrange the order of the result as you suggested.

3) There are many grammatical mistakes that we did not list in this round because of the major issues identified. Hopefully the authors can get some help with this in the future. It might be a good idea to employ professional editing services or at least AI-based tools that detect problems and suggest improvements such as Grammarly. We are aware that the authors are not native speakers and they should not be overly criticized for these issues.

> Thank you for your suggestion. We corrected all the grammatical errors we found using Grammarly.

4) Line 63: "... have also revealed the filamentous structures in the R conformation" – this is not correct: the structure was pseudo-filamentous as we commented on as reviewers at the time - and the authors (some the same here) agreed.

> We replaced the word "filamentous" with "pseudo-filamentous".

5) Line 70: we think that the sentence is overstating what is still missing: the crystal structures in the T / closed form are pretty accurate as the authors comment in the present

manuscript.

> We intended that no high-resolution structures are available if limited to the solution structures. We modified the sentence as follows, “However, due to the lack of high-resolution solution structures of the FtsZ protofilament, the relationships between these different types of polymers, the monomeric T/R conformations, and the bound nucleotides remain to be elucidated.”

6) Supp Figures 9 and 10: We are not convinced that the density is good enough. Especially Figure S10 reveals that the density is blobby and does not describe the nucleotide well at all. Is it possible that the reconstruction suffered from preferred orientation bias? Could you please show the orientation distribution of the particles that went into the final reconstruction? Wagstaff et al reported very similar problems (Wagstaff et al Sci Adv [2023]). Also needed: density close-ups, anisotropy analysis and feature-based resolution estimation.

> We thought that the previous Supplementary Fig. 10b was misleading, because only the map at high contour level was shown to emphasize the strong density corresponding to the three phosphates in the GMPCPP. Now we added the same map to the new Fig. 2f and overlaid it with the lower-level contoured map. We also added the orientation distribution of the particles used in the final reconstruction, FSC curves and the sphericity calculated by the 3DFSC server, and density close-ups to Supplementary Fig. 2 and the resolution estimation from the model ($d_{\text{model}} = 3.5 \text{ \AA}$) to Supplementary Table 1.

7) Line 266: see last point: we are not convinced that the GMPCPP density in the cryo-EM structure justifies this statement.

> We deleted the following sentence, “However, the binding modes of the GTP analogs were different.” as the density is not clear enough to determine the accurate position and conformation of the GMPCPP. We also deleted our claim about the intermediate of GTP hydrolysis. However, as described above, there is a strong density that can accommodate three phosphates directed toward the T7 loop. We think this result indicates that such kind of GMPCPP conformation is contained in the T protofilaments as a major component.

8) Line 325 and below: we disagree with the idea that the ring forms because FtsZ hydrolyses GTP to GDP and then curves. As the text says, GDP FtsZ depolymerizes and is not part of the ring. This drives treadmilling instead.

Some editorial comments, in case they help, since it is fully understood that the authors are not native speakers:

> We deleted this part and added the sentences about the relationship between our structures and the cytomotive switch model, as described above.

9) The title misses an article we think: "Structures of a single protofilament of FtsZ and a double-helical tube of an FtsZ-monobody complex". Maybe better: "Structures of a single protofilament of FtsZ and in complex with a monobody"

>Thank you for your suggestion. We changed the title to "Structures of a FtsZ single protofilament and a double-helical tube in complex with a monobody".

10) Introduction: Rather than "In this study, based on the tenth human fibronectin type III domain (FN3), we developed a monobody (Mb), a synthetic protein that binds to both KpFtsZ and EcFtsZ at high affinity, to aid high- resolution structure determination of the FtsZ protofilament", have something like "In order to aid high-resolution structure determination of the FtsZ protofilament, we developed a monobody (Mb) that binds to both KpFtsZ and EcFtsZ with high affinity." There is no need for a detailed explanation of what a monobody is. Also, "with" rather than "at". "at" makes it sound as if the authors are describing the conditions under which binding occurs.

> We modified the sentence as you suggested.

11) Results: "the orientation of the molecule is slightly tilted with each other." What? Maybe "the relative orientation of the molecules is slightly different."?

> We modified the expression.

12) Discussion: May we suggest in the discussion, instead of "The difference in the polymer morphology formed in the presence of two GTP analogs, GMPPNP for the double-helical tube and GMPCPP for the single protofilament, can be explained by the fact that GMPPNP did not replace endogenous GDP as revealed in the structure of the tube with bound GDP despite the addition of GMPPNP in solution (Fig. 4g), which is possibly because of the lower affinity of GMPPNP for FtsZ than GDP. Notably, FtsZ always binds endogenous GDP brought in from E. coli cells even without adding nucleotides, as we have experienced in our previous crystal structure analyses." to have "We have experienced in our previous crystal structure analyses that without added nucleotides, FtsZ binds endogenous GDP from E. coli cells. The structure of our tube reveals that, despite the addition of GMPPNP in solution (Fig. 4g), endogenous GDP was not replaced. This may be because GMPPNP's affinity for FtsZ is lower than GDP's." Different polymer morphologies seem to form depending the nucleotide or nucleotide

analog bound: double-helical tubes in the case of GMPPNP and single protofilaments for GDP." The wording as it stands now seems to say that the tube structure is what FtsZ would do in the presence of GMPPNP, when the authors want to say that the tube structure has bound GDP.

> We modified the sentences as you suggested.

13) The authors have used "intimate" a couple of times. This is perhaps a bit too poetic: "close" is just fine.

> We replaced "more intimate" with "closer" throughout the manuscript.

14) Change "stabilized the tubular structure of KpFtsZ in solution (Fig. 3) for its high-resolution cryoEM study (Fig. 4)." to "stabilized the tubular structure of KpFtsZ in solution (Fig. 3) for its study by high-resolution cryo-EM (Fig. 4)."

> Thank you for your suggestion, but we deleted this part as we revised the discussion.

15) In the discussion, the authors write "Now that KpFtsZ can be in either of the T and R conformations depending on the bound nucleotide in different polymer structures formed in solution, not only for the intramolecular structure but also for the intermolecular interactions along the protofilament with different curvatures, there is no doubt that the roles and mechanisms of structural transition of FtsZ between the T and R conformations are universally shared by a wide range of species and is canonical." We think that the authors' claims are a bit too bold. They may be right, but what they have shown here does not justify language like "no doubt" and "universally". We would also argue strongly that the structures solved do not support the idea that nucleotide state controls the closed / R to open / T transition, given that monobodies are involved and all but one structure are constraint by crystal contacts (as the authors state themselves). There is no open / T structure that is monomeric, for example, but there are closed / R structures in all nucleotide states and they are all monomeric (or at least not properly polymeric with the T7 loop able to activate hydrolysis).

> We deleted this part and added the following sentences instead, "Now we confirm that the solution structure of KpFtsZ single protofilament adopts the T conformation in the presence of GMPCPP, in contrast to the crystal structures. Our results may support the cytomotive switch model, where the conformation switch between the T and R states is induced by the intermolecular interactions during polymerization and depolymerization and not by the changes in nucleotide states. The ambiguity of the cryoEM map corresponding to the GMPCPP and the T7 loop might indicate that they are in a mixture

of multiple conformations in the protofilaments with the T conformation. Conversely, the T conformation of FtsZ in a protofilament may not be strongly affected by the nucleotide states and the conformations of the T7 loop as far as FtsZ polymerizes.”